# Coordinated calcium signalling in cochlear sensory and non-sensory cells refines afferent innervation of outer hair cells

Federico Ceriani[1,†] iD, Aenea Hendry[1,†], Jing-Yi Jeng[1], Stuart L Johnson[1], Friederike Stephani[2], Jennifer Olt[1], Matthew C Holley[1], Fabio Mammano[3,4] iD, Jutta Engel[2], Corné J Kros[5], Dwayne D Simmons[6] & Walter Marcotti[1,*] iD

## Abstract

Outer hair cells (OHCs) are highly specialized sensory cells conferring the fine-tuning and high sensitivity of the mammalian cochlea to acoustic stimuli. Here, by genetically manipulating spontaneous $Ca^{2+}$ signalling in mice *in vivo*, through a period of early postnatal development, we find that the refinement of OHC afferent innervation is regulated by complementary spontaneous $Ca^{2+}$ signals originating in OHCs and non-sensory cells. OHCs fire spontaneous $Ca^{2+}$ action potentials during a narrow period of neonatal development. Simultaneously, waves of $Ca^{2+}$ activity in the non-sensory cells of the greater epithelial ridge cause, via ATP-induced activation of $P2X_3$ receptors, the increase and synchronization of the $Ca^{2+}$ activity in nearby OHCs. This synchronization is required for the refinement of their immature afferent innervation. In the absence of connexin channels, $Ca^{2+}$ waves are impaired, leading to a reduction in the number of ribbon synapses and afferent fibres on OHCs. We propose that the correct maturation of the afferent connectivity of OHCs requires experience-independent $Ca^{2+}$ signals from sensory and non-sensory cells.

**Keywords** calcium waves; hair cells; pre-hearing development; purinergic receptors; spontaneous activity
**Subject Categories** Development & Differentiation; Membrane & Intracellular Transport
The EMBO Journal (2019) 38: e99839

See also: **MG Leitner & D Oliver** (May 2019)

## Introduction

Mammalian hearing depends upon two specialized sensory receptor cell types in the organ of Corti, the inner and outer hair cells, and their afferent and efferent neuronal connections. The differentiation, maturation and maintenance of these neuronal connections require precise timing and coordination between genetic programmes and physiological activity (Corns *et al*, 2014, 2018; Delacroix & Malgrange, 2015). Inner hair cells (IHCs) are the primary sensory receptor cells, and they relay sound information to spiral ganglion afferent neurons via the release of glutamate from vesicles tethered to pre-synaptic ribbons. By contrast, the role of outer hair cells (OHCs) is to extend the functional dynamic range of the mammalian cochlea and to enhance the sensitivity and the frequency tuning within the cochlear partition (Dallos, 1992). Adult OHCs are primarily innervated by cholinergic medial olivocochlear neurons (Liberman, 1980; Maison *et al*, 2003), the role of which is to modulate mechanical amplification in the adult cochlea (Guinan, 1996). However, OHCs are also innervated by type II afferent fibres that appear to be activated by acoustic trauma (Flores *et al*, 2015; Liu *et al*, 2015), unlike the type I fibres contacting IHCs that encode sound timing, intensity and frequency. In most altricial rodents, OHCs only begin to acquire the innervation pattern present in the mature cochlea towards the end of the first and the start of the second postnatal week (Simmons, 1994; Simmons *et al*, 1996). However, the molecular mechanisms responsible for the correct afferent innervation of OHCs remain poorly understood.

The refinement of sensory circuits during development is normally influenced by periods of experience-independent action potential (AP) activity before the onset of function (Katz & Shatz, 1996; Blankenship & Feller, 2010). Calcium-dependent APs have been shown to occur spontaneously in immature IHCs (Johnson

1  Department of Biomedical Science, University of Sheffield, Sheffield, UK
2  Center for Integrative Physiology and Molecular Medicine (CIPMM), Saarland University, Homburg, Germany
3  Department of Physics and Astronomy "G. Galilei", University of Padua, Padova, Italy
4  Department of Biomedical Sciences, Institute of Cell Biology and Neurobiology, Italian National Research Council, Monterotondo, Italy
5  School of Life Sciences, University of Sussex, Brighton, UK
6  Department of Biology, Baylor University, Waco, TX, USA
    *Corresponding author. Tel: +44 114 2221098; E-mail: w.marcotti@sheffield.ac.uk
    † These authors contributed equally to the work

*et al*, 2011, 2017) but not in OHCs (Oliver *et al*, 1997; Marcotti & Kros, 1999; Weisz *et al*, 2012). One study reported spontaneous APs in OHCs of wild-type and otoferlin mutant mice, but mostly using elevated extracellular Ca$^{2+}$ and high intracellular EGTA (Beurg *et al*, 2008).

We found that during a narrow, critical period of postnatal development (around birth), OHCs show spontaneous Ca$^{2+}$ signals immediately preceding their functional maturation at ~ P7–P8. This Ca$^{2+}$ activity in immature OHCs can be modulated by Ca$^{2+}$ waves travelling among non-sensory cells via the ATP-dependent activation of P2X$_3$ receptors. The Ca$^{2+}$ waves, by increasing the Ca$^{2+}$ signals in OHCs, were able to synchronize the activity of nearby OHCs. The reduction of spontaneous Ca$^{2+}$ waves in non-sensory cells *in vivo* prevented the maturation of the OHC afferent innervation. We propose that precisely modulated Ca$^{2+}$ signals between OHCs and non-sensory cells are necessary for the correct maturation of the neuronal connectivity to OHCs.

# Results

The functional development of OHCs was studied primarily in the apical third of the mouse cochlea, corresponding to a frequency range in the adult mouse of ~ 6–12 kHz (Müller *et al*, 2005; Fig 1A). For comparison, some recordings were also made from the basal coil of the cochlea through the frequency range of ~ 25–45 kHz (Fig 1A). Spontaneous Ca$^{2+}$ activity in immature OHCs and its modulation by non-sensory cells in the greater (GER) and lesser (LER) epithelial ridges (Fig 1B) was recorded from cochleae bathed in a perilymph-like extracellular solution (1.3 mM Ca$^{2+}$ and 5.8 mM K$^+$; Wangemann & Schacht, 1996) either near body temperature or at room temperature. Although the stereociliary bundles of hair cells (Fig 1B) are normally bathed in endolymph, which contains ~ 150 mM K$^+$ and ~ 20 μM Ca$^{2+}$ in the mature cochlea (Bosher & Warren, 1978; Wangemann & Schacht, 1996), during the first few days after birth endolymph has a similar ionic composition to that of the perilymph (Wangemann & Schacht, 1996).

## Calcium-dependent activity in OHCs occurs spontaneously during a narrow period of development

Spontaneous, rapid Ca$^{2+}$ transients were recorded from OHCs maintained at near-body (~ 35°C: Fig 1C, Movie EV1) and room temperature (~ 20°C: Fig 1D, Movie EV3, top panel) in acutely dissected cochleae from newborn mice loaded with the Ca$^{2+}$ indicator Fluo-4. Similar Ca$^{2+}$ transients were observed in OHCs in 37 separate recordings from 13 different mice. By combining cell-attached patch clamp recordings and Ca$^{2+}$ imaging, we confirmed that the Ca$^{2+}$ signals represent the optical readout of OHC firing activity, with bursts of APs causing large increases in the OHC Ca$^{2+}$ level (Fig 1E). Although Ca$^{2+}$ signals were present in OHCs along the entire cochlea at birth, the number of cells showing this activity decreased over time, with basal OHCs being the first to stop at around P4 and apical cells stopping a couple of days later (Fig 1F). This correlated with a decrease in the maximum Ca$^{2+}$-related change in fluorescence intensity ($\Delta F/F_0$) for Ca$^{2+}$ measured from active OHCs (Fig 1G), which is likely due to a progressive disappearance of Ca$^{2+}$ activity caused by the reduction of the Ca$^{2+}$

current (Knirsch *et al*, 2007) and upregulation of the K$^+$ currents (Marcotti & Kros, 1999) in OHCs with age.

Calcium transients were abolished in Ca$^{2+}$-free solution (Appendix Fig S1A–C; Movie EV2) and were absent in OHCs lacking the Ca$_V$1.3 Ca$^{2+}$ channel subunit (Appendix Fig S1D; Movie EV3, bottom panel), the main voltage-gated Ca$^{2+}$ channel expressed in hair cells (Platzer *et al*, 2000; Michna *et al*, 2003). These results, together with the finding that Ca$^{2+}$ activity was not prevented when blocking Ca$^{2+}$ release from intracellular stores (Appendix Fig S1E), indicate their dependence on extracellular Ca$^{2+}$.

## Calcium waves from non-sensory cells coordinate OHC Ca$^{2+}$ signals

Hair cells are embedded in a matrix of non-sensory, epithelial supporting cells (Fig 1B). The inner phalangeal cells surrounding the IHCs and the tightly packed columnar cells that form Kölliker's organ are part of the GER, and they show spontaneous inward currents (Tritsch *et al*, 2007). This spontaneous activity is initiated by extracellular ATP, which is released via an extensive network of connexin hemichannels in non-sensory cells, activating purinergic autoreceptors on the same cells, causing an increase in intracellular Ca$^{2+}$. It leads to spatially and temporally coordinated Ca$^{2+}$ waves that are propagated across the epithelium (Tritsch *et al*, 2007). Calcium waves can also be triggered by the application of ATP to non-sensory cells surrounding the OHCs in the LER (e.g. Deiters' cells: see Fig 1B), but these waves are thought not to occur spontaneously (Tritsch *et al*, 2007; Anselmi *et al*, 2008). Therefore, we investigated whether spontaneous Ca$^{2+}$ waves originating in the GER of the developing mouse cochlea (Fig 2A, red arrow) can influence the Ca$^{2+}$ activity in OHCs. In the proximity of large Ca$^{2+}$ waves, OHCs showed an increased Ca$^{2+}$ activity (Fig 2B, see also Fig 2G and H), which was most likely driven by their depolarization and subsequently increased AP firing rate (Fig 1E).

The increased OHC Ca$^{2+}$ signals followed very closely the time course of the Ca$^{2+}$ wave originating in the GER. As a consequence of the increased OHC Ca$^{2+}$ signals, the otherwise uncorrelated spontaneous Ca$^{2+}$ activity in nearby OHCs became highly correlated temporally during large Ca$^{2+}$ waves (Fig 2B and D, Movie EV4). However, OHC Ca$^{2+}$ signals remained uncorrelated in the absence of waves or during small Ca$^{2+}$ waves in the GER (Fig 2C and E, Movie EV5). This suggests that Ca$^{2+}$ waves may serve as an extrinsic pathway to coordinate the firing activity of otherwise independent nearby OHCs. To quantify the change in the synchronization of the OHC Ca$^{2+}$ signals, we computed the average pairwise correlation coefficient ($r_s^{avg}$: see Materials and Methods). This correlation coefficient was measured between every pair of OHCs in the field of view (64 ± 5 OHCs, seven cochleae, six mice) during a time window of 13.2 s (400 frames, grey area in Fig 2B and C, right panels) centred on the maximum intensity of the spontaneous Ca$^{2+}$ signal occurring in the GER. The average correlation coefficient in nearby OHCs showed a positive relationship with the longitudinal (i.e. along the tonotopic axis) extension of Ca$^{2+}$ waves in the GER (Pearson's correlation coefficient: 0.85, Fig 2F). While half of the smaller Ca$^{2+}$ waves that spread over less than 75 μm (28 out of 56) had no significant effect on the correlation, all 19 of the larger waves analysed were able to synchronize the activity of several OHCs (Fig 2F).

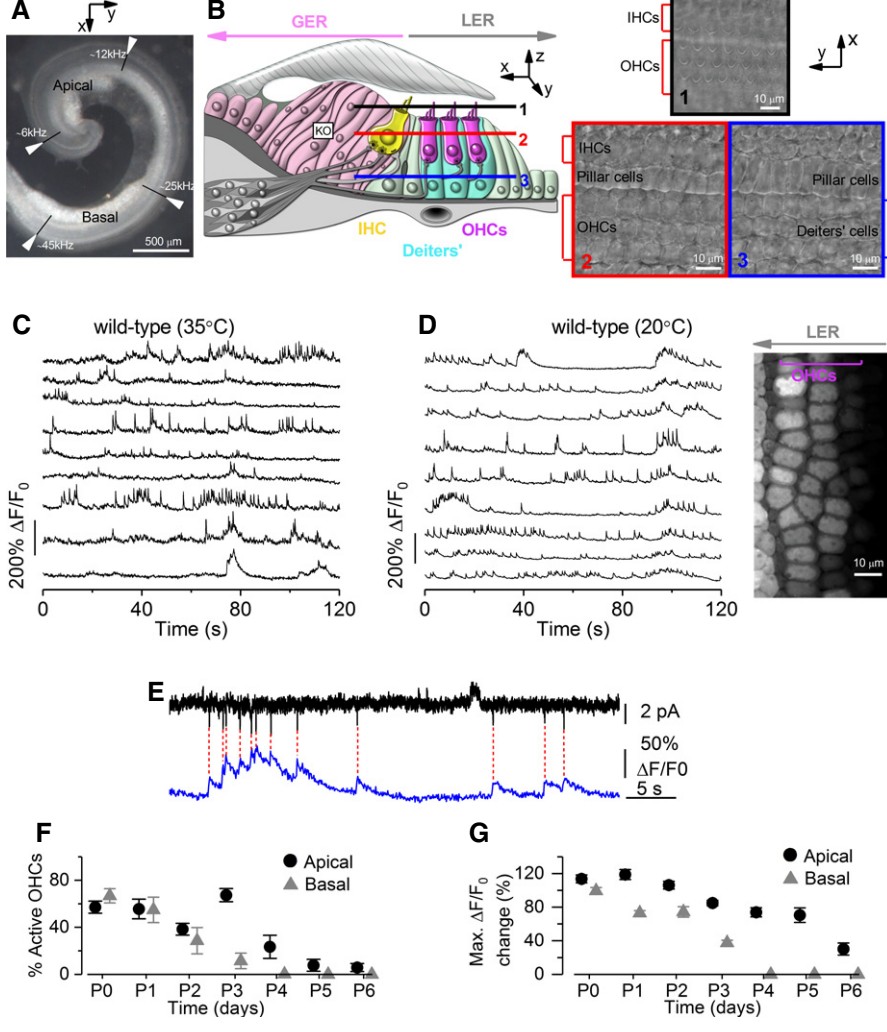

**Figure 1. Early postnatal OHCs show spontaneous Ca²⁺ signals.**

A     Image of the mouse cochlea at P2 highlighting the apical and basal regions used for the experiments. The apical and basal regions were at a fractional distance along the coil of approximately 8–32% (corresponding to a frequency range in the mature mouse of ~ 6–12 kHz) and 55–80% (~ 25–45 kHz) from the apex, respectively.

B     Diagram (left) showing a cross-section of an early postnatal organ of Corti. OHCs: outer hair cells; IHCs: inner hair cells; GER: greater epithelial ridge, which includes non-sensory cells such as the inner phalangeal cells (surrounding the IHCs) and tightly packed tall columnar cells forming the Kölliker's organ (KO); LER: lesser epithelial ridge. Right panels show DIC images of the cochlea at the level of the hair bundle (top) and both OHCs and non-sensory Deiters' cells in the LER region (bottom).

C, D     Representative ΔF/F₀ traces from nine apical OHCs of a P2 wild-type mouse (from the image in the right panel) recorded at body (C) and room (D) temperature. Traces are computed as pixel averages of regions of interest centred on OHCs. Calcium signals are evident in OHCs. In this and the following panels, the right image provides a visual representation of the spontaneous activity over the entire duration of the recordings (120 s) and was obtained by averaging 4,000 frames of raw data.

E     Simultaneous cell-attached patch clamp recording (top) and Ca²⁺ imaging (bottom) obtained from a P1 OHC from wild-type mouse at room temperature. The intracellular Ca²⁺ level in OHCs increased rapidly but decayed with a relatively long fluorescence decay time constant (~ 300 ms; Ceriani *et al*, 2016).

F     Percentage of apical and basal OHCs showing spontaneous Ca²⁺ signals at near body temperature and as a function of postnatal age. Note that these values likely represent an underestimation of the fraction of active OHCs, since the three-dimensional structure of the cochlea and the optical sectioning capability of 2-photon microscopy make it difficult to be in the optimal focal conditions for the simultaneous recording of all three rows of OHCs. Moreover, active OHCs were those showing activity within the 2-min recording time. Number of total OHCs and recordings from left to right were as follows: apical cochlea 384 and 8; 338 and 7; 441 and 10; 497 and 9; 466 and 10; 453 and 11; 254 and 6; and basal cochlea 408 and 9; 528 and 11; 460 and 10; 380 and 7; 676 and 12; 143 and 3; 133 and 4. Values are mean ± SEM.

G     Maximum ΔF/F₀ changes in apical and basal active OHCs as a function of postnatal age. Number of active OHCs were as follows: apical cochlea 218; 189; 82; 336; 94; 39; 6; and basal cochlea 273; 303; 124; 46; 0; 0; 0. These OHCs came from the dataset in (F). Values are mean ± SEM.

In order to provide an estimate of the increased Ca²⁺ signal in OHCs during the Ca²⁺ waves from non-sensory cells, we quantified the time integral of the fluorescence traces recorded from OHCs (see Materials and Methods). We found that OHCs closer to the Ca²⁺ wave showed a larger increase in Ca²⁺ activity compared to those located far away (Fig 2G). We then compared the increased Ca²⁺

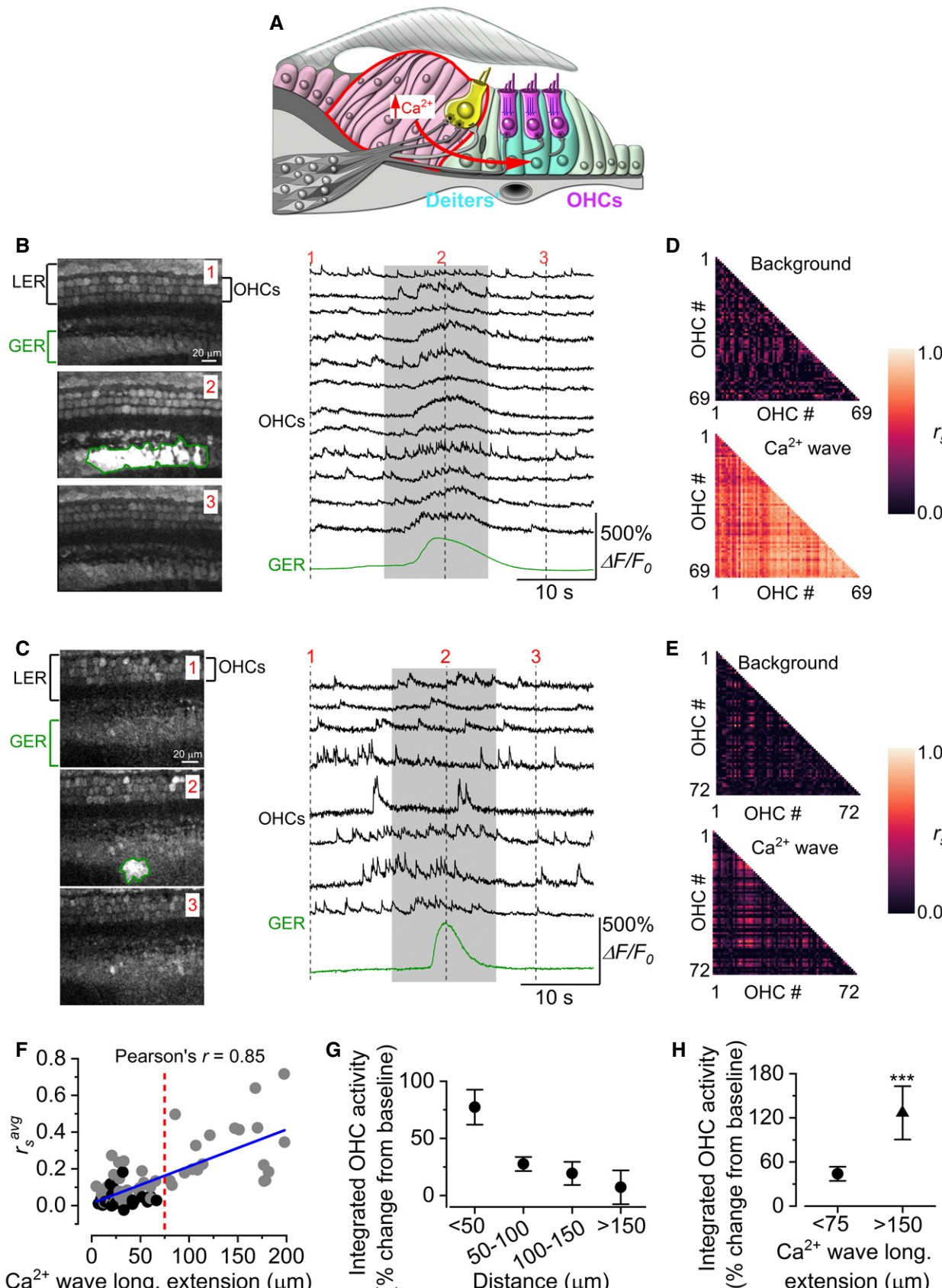

**Figure 2.**

**Figure 2.  Calcium waves from the GER modulate OHC spontaneous Ca²⁺ signalling.**

A  Diagram showing a cross-section of an immature organ of Corti. Spontaneous Ca²⁺ waves (curved red arrow) are generated in the non-sensory cells present in the greater epithelial ridge (GER: red line).

B, C  Three representative images (left panels) obtained before (1), during (2) and after (3) the spontaneous appearance of a wide (B) and a narrow (C) Ca²⁺ wave in the GER in the apical coils of P2 wild-type mice. Right panels show representative $\Delta F/F_0$ traces from 12 (B) and 8 (C) OHCs (black traces) and those originating from the Ca²⁺ wave in the GER (green traces). The grey-shaded areas highlight the time window used for correlation analysis (see below). Recordings were made at 31°C.

D, E  Correlation matrices computed from the Ca²⁺ fluorescence traces of 69 (D) and 72 (E) OHCs from panel (B) (large Ca²⁺ wave) and panel (C) (small Ca²⁺ wave). Correlation coefficients were computed before (top panels: background) and during (bottom panels: Ca²⁺ wave) the occurrence of the Ca²⁺ wave in nearby non-sensory cells. Each matrix element represents the Spearman's rank correlation coefficient ($r_s$: see Materials and Methods) of one pair of OHCs.

F  Average Spearman's rank correlation coefficient ($r_s^{avg}$: see Materials and Methods) between the Ca²⁺ activity in OHCs as a function of the longitudinal extension of spontaneous Ca²⁺ waves in the GER from the apical coil of P1–P2 mouse cochleae. The average length of the apical-coil segments used for these experiments was $188 \pm 4$ μm (see Materials and Methods). Grey dots represent Ca²⁺ waves that are associated with a significant increase in OHC correlation, while black dots represent events during which OHC correlation did not increase significantly. Waves travelling more than 75 μm in the longitudinal direction (red dashed line) always triggered a significant increase in OHC synchronization. Solid line in panel (F) represents a linear fit to the data. The slope was $(2.60 \pm 0.26)10^{-3}$ μm⁻¹, significantly different from zero ($P < 0.0001$; ANOVA, *f*-test).

G  Fractional increase in the integral of the fluorescence Ca²⁺ traces recorded from OHCs as a function of their distance from the Ca²⁺ waves. Overall one-way ANOVA: $P = 0.0007$. Number of recordings from left to right: 18, 19, 19, 7.

H  Average increase in the integral of the Ca²⁺ traces in OHCs for small (< 75 μm) and large (> 150 μm) Ca²⁺ waves. Number of recordings: < 75 μm 56; < 150 μm 8. These two ranges were selected to emphasize the effect of the different extension of the Ca²⁺ waves recorded.

signals in OHCs positioned at around 100 μm from the Ca²⁺ waves that spread over < 75 μm with those that spread over > 150 μm (Fig. 2H), and found that larger Ca²⁺ waves caused a significantly increased Ca²⁺ activity in OHCs ($P = 0.0054$, *t*-test). We also found that both the increased OHC Ca²⁺ activity and the degree of correlation ($r_s^{avg}$) were independent from the amplitude ($\Delta F/F_0$) of the Ca²⁺ waves (Appendix Fig S2A and B). Therefore, the coordination of the Ca²⁺ signals between nearby OHCs was dependent on the lateral spread, but not the amplitude, of the Ca²⁺ waves.

We then sought to identify how spontaneous Ca²⁺ activity from the non-sensory cells of the GER coupled to Ca²⁺ signalling in OHCs. Patch clamp recordings from Deiters' cells, which surround the OHCs in the LER (Fig 2A), revealed spontaneous inward currents similar to those measured in non-sensory cells of the GER (Tritsch *et al*, 2007). During these spontaneous currents, Deiters' cells depolarized by $16.7 \pm 0.5$ mV (range 5.4–43.5 mV, 339 events, $n = 9$) from an average resting membrane potential of $-70.8 \pm 1.6$ mV ($n = 9$; Fig 3A and B). Simultaneous recordings showed that the inward currents in the Deiters' cells appear synchronized with Ca²⁺ waves in the GER in both wild-type (Appendix Fig S3A) and $Ca_V1.3^{-/-}$ mice (Fig 3C). The absence of Ca²⁺ signals in OHCs from $Ca_V1.3^{-/-}$ mice made it easier to see that the Ca²⁺ waves originating in the GER were able to travel to the LER and propagate through Deiters' cells (Fig 3D; Appendix Fig S3B). Calcium waves originating in the GER reached the more distant Deiters' cell in the radial direction with a delay of $1.67 \pm 0.55$ s ($n = 11$ recordings, 5 cochleae, 3 mice). In order to test whether Deiters' cells mediate signal transfer from the GER to the OHCs, we analysed Ca²⁺ signals after removing a few Deiters' cells beneath the area of interest (Fig 4C) using gentle suction via a small pipette (~ 3–4 μm in diameter). This procedure is widely used to gain access to the different cochlear cell types, including the OHCs (Marcotti & Kros, 1999). Importantly, this procedure does not affect OHC integrity, since they retained normal biophysical characteristics (e.g. resting membrane potential and ability to fire action potentials: Appendix Fig S4A–G), the ability to generate Ca²⁺ transients (Appendix Fig S4H) and sensitivity to extracellular ATP (Appendix Fig S4I). After the removal of the Deiters' cells, we elicited Ca²⁺ waves by photo-damaging a small area of the GER at room temperature. This treatment was used as a proxy for

spontaneous Ca²⁺ waves since they share the same connexin- and ATP-dependent molecular mechanism (Gale *et al*, 2004; Tritsch *et al*, 2007; Lahne & Gale, 2010), while allowing precise temporal and spatial control of Ca²⁺ wave occurrence. Moreover, spontaneous (Appendix Fig S5) and photo-damage-induced Ca²⁺ waves (Fig 4A and B, Movie EV6) have a qualitatively similar influence on OHC Ca²⁺ activity. We found that Ca²⁺ elevation in OHCs associated with induced Ca²⁺ waves in the GER was almost completely abolished when the nearby Deiters' cells were removed (Fig 4C–E: $P < 0.001$ compared to when Deiters' cells were present; post-test from one-way ANOVA) or in $Ca_V1.3^{-/-}$ mice (Fig 4C–E). We conclude that Deiters' cells are essential intermediaries for coupling activity from the GER to OHCs.

## ATP triggers OHC Ca²⁺ signals in the developing cochlea

To determine the molecular mechanism linking activity in the Deiters' cells with OHC synchronization, we pharmacologically probed the basolateral membrane of OHCs deprived of their surrounding Deiters' cells (Fig 5A). Deiters' cells release ATP via connexin hemichannels (Zhao *et al*, 2005), and immature OHCs exhibit depolarizing, ATP-gated currents (Glowatzki *et al*, 1997). In the absence of Deiters' cells, we found that local perfusion of 10 μM ATP onto the basolateral membrane of OHCs triggered large Ca²⁺ responses (Fig 5B). The OHC response to ATP was abolished in $Ca_V1.3^{-/-}$ mice even at 100 μM (Appendix Fig S6, Movie EV7). Under whole-cell patch clamp, 10 and 100 μM ATP caused OHCs to depolarize by $15.8 \pm 2.9$ mV (steady-state, $n = 10$, P1–P2; Fig 5C; Appendix Fig S6). Extracellular ATP can act on ionotropic (P2X) and metabotropic (P2Y) purinergic receptors, both of which are present in cochlear hair cells (Housley *et al*, 2006). We found that Ca²⁺ signals from OHCs were either abolished or greatly reduced when ATP was applied together with the purinergic receptor antagonists, suramin (200 μM: Fig 5D and I) and PPADS (Fig 5I). Under the whole-cell patch clamp configuration, suramin reduced the ATP responses by $89.8 \pm 6.5\%$ ($n = 4$, P1; Fig 5C). The absence of ATP-induced Ca²⁺ signals in wild-type OHCs bathed in a Ca²⁺-free solution (Fig 5E and I) and in $Ca_V1.3^{-/-}$ mice (Fig 5I, Appendix Fig S6, Movie EV7) indicates that P2Y receptors, which mobilize Ca²⁺ from intracellular stores, are unlikely to be involved in mediating OHC responses to

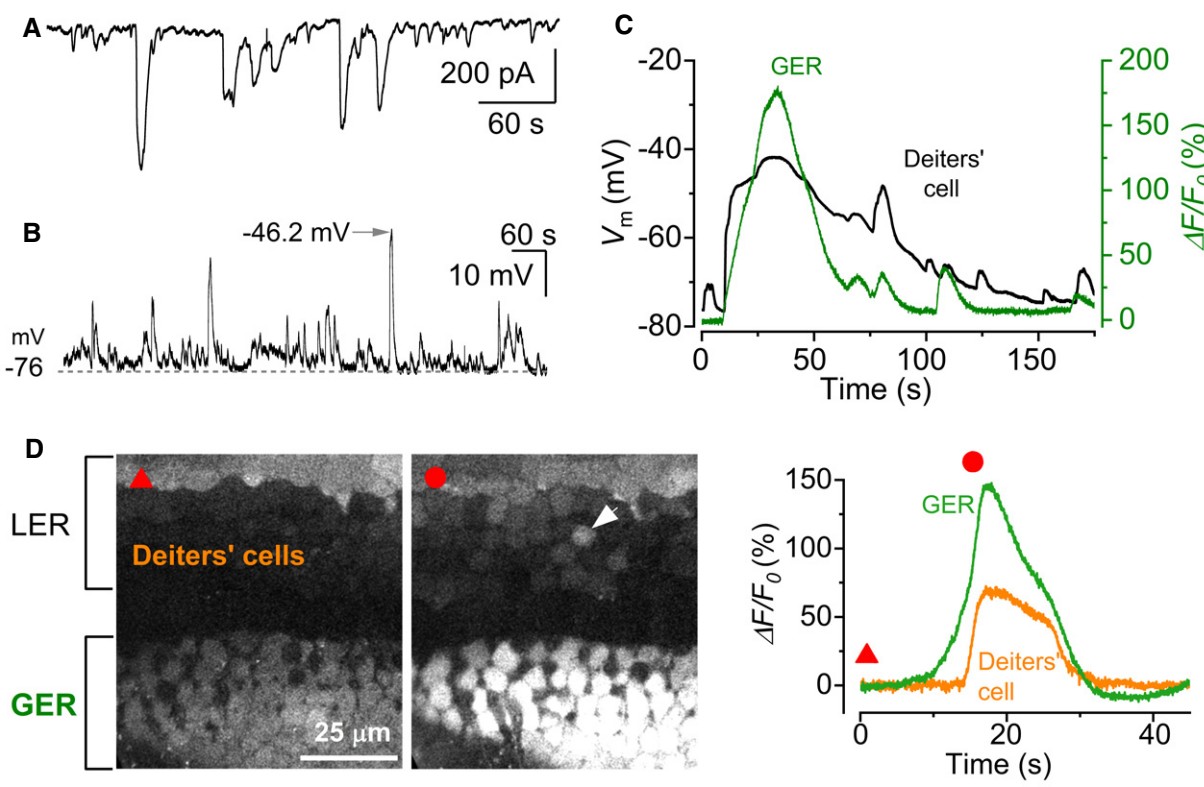

**Figure 3. Calcium signalling from the GER travels to Deiters' cells in the LER.**

A, B   Spontaneous activity recorded using the patch clamp technique from Deiters' cells using whole-cell voltage clamp (A) and current clamp (B) in a P1 mouse cochlea. Recordings were made at RT.

C   Simultaneous recording of whole-cell voltage responses in a Deiters' cell in the LER (black trace) and a spontaneous Ca²⁺ wave from the GER ($\Delta F/F_0$) of a $Ca_V 1.3^{-/-}$ P1 mouse (for wild-type mouse cochlea, see Appendix Fig S1).

D   Representative $\Delta F/F_0$ traces (right traces) from the GER and one Deiters' cell (arrowhead in the middle panel) of a P1 $Ca_V 1.3^{-/-}$ mouse. The red triangle (left panel) shows the fluorescence level before the Ca²⁺ wave, and the red circle (middle panel) shows the response during the wave. Note that the response in the GER occurs earlier than that in the Deiters' cell (left), even though the rapid onset of the voltage change in the Deiters' cell (C) gives the impression of an overlapping event. Traces are computed as pixel averages from a single non-sensory cell in the GER and a Deiters' cell positioned in the radial direction to the Ca²⁺ wave. The onset of the Ca²⁺ waves was defined by a fluorescence increase of five times the standard deviation of the background noise compared to baseline.

ATP at this developmental stage (King & Townsend-Nicholson, 2003; Egan & Khakh, 2004). Consistent with this hypothesis, the application of the phospholipase C inhibitor U73122, which prevents the IP₃-mediated Ca²⁺ release from intracellular Ca²⁺ stores linked to the activation of P2Y receptors (Bleasdale & Fisher, 1993; Lahne & Gale, 2010), did not inhibit ATP-induced Ca²⁺ signals in OHCs (Fig 5F and I). The local application of 1 μM UTP, a selective agonist of P2Y receptors that mobilizes Ca²⁺ from intracellular stores in cochlear non-sensory cells (Piazza *et al*, 2007), onto Deiters' cells triggered an increase in their intracellular Ca²⁺ levels, followed by increased Ca²⁺ activity in nearby OHCs (Fig 5G, Movie EV8). When the Deiters' cells were removed, OHCs were not affected by UTP (Fig 5H: normalized maximal response: DCs intact: $1.00 \pm 0.23$, $n = 4$ recordings, 3 cochleae, 3 mice; DCs removed: $0.20 \pm 0.04$, $n = 5$ recordings, 3 cochleae, 3 mice; $P < 0.01$, Mann–Whitney *U*-test), providing further evidence that ATP-induced Ca²⁺ signals from these non-sensory cells directly modulate OHC activity. Altogether these data indicate that ATP, acting through P2Y receptors in Deiters' cells and P2X receptors in OHCs, can coordinate the activity of nearby OHCs.

We then sought to investigate whether the purinergic signalling from Deiters' cells to OHCs during Ca²⁺ waves was also present in the intact cochlear preparation (i.e. without removing the Deiters' cells as done for Fig 5A). We initially investigated the Deiters' cell to OHC coupling while applying the non-selective purinergic receptor antagonist PPADS. Even though the occurrence of Ca²⁺ waves was reduced in the presence of PPADS (control: $2.09 \pm 0.21$ events min⁻¹, $n = 32$ recordings, 7 cochleae, 6 mice, P1–P2; PPADS: $0.81 \pm 0.14$ events min⁻¹, $n = 16$ recordings, 11 cochleae, 7 mice, P1–P2, $P < 0.0001$, Mann–Whitney *U*-test), large Ca²⁺ waves originating in the GER were still able to reach the LER, but failed to increase, and as such synchronize, the Ca²⁺ signals in OHCs (Fig 6A,C and E). Of the other known P2X receptors present in the cochlea, P2X₃ has been shown to be transiently expressed during early stages of development (Huang *et al*, 2006). We found that the specific P2X₃ antagonist A317491 (Jarvis *et al*, 2004) was able to prevent the Ca²⁺ waves from affecting the Ca²⁺ signals in OHCs (Fig 6B,D and F). We further tested the presence of P2X₃ receptors in OHCs by performing current-clamp recordings and found that A317491 fully and reversibly blocked ATP-induced OHC

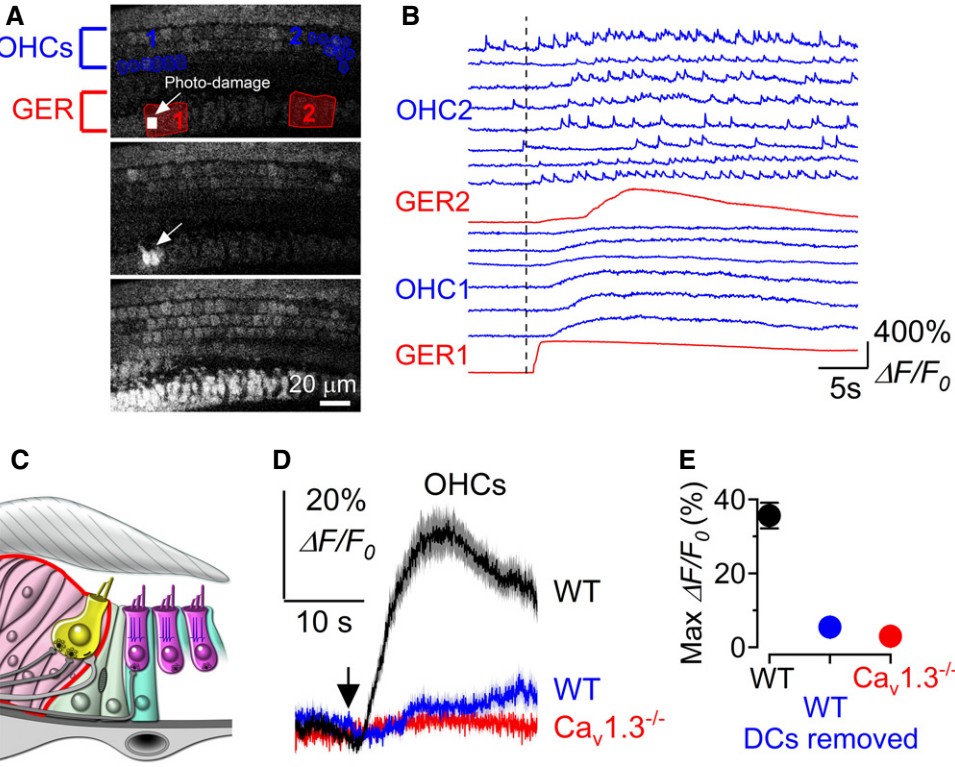

**Figure 4.  Deiters' cells are required for the modulation of OHC Ca²⁺ activity by Ca²⁺ signalling from the GER.**

A     Three representative images showing the small photo-damaged region in the GER (top panel, small white rectangles and arrow), the initiation of the Ca²⁺ wave from the damaged non-sensory cells (middle panel: arrow) and the full extent of the Ca²⁺ wave (bottom panel). Note the increase in fluorescence of OHCs during Ca²⁺ wave. The red regions of interest in the top panel are used to measure Ca²⁺ signals at the point where the Ca²⁺ wave was generated (GER1) and its propagation along the epithelium (GER2). The blue ROIs are those used to measure OHC Ca²⁺ signalling. Recording was performed at RT from wild-type P2 mouse. The photo-damage area was 55 μm² in size (18 × 23 pixels) and typically covered the apical surface of ~ 1–2 non-sensory cells in the GER.

B     Representative $\Delta F/F_0$ traces from the GER (red traces) and OHCs (blue traces). Note that the six OHCs traces named "OHC1" are those closer to GER1 (photo-damage region: see top panel A), while the eight OHC traces named "OHC2" are those near GER2 (far away from the photo-damage region: see top panel A). Recordings were made at RT.

C     Diagram showing the immature organ of Corti without the Deiters' cells. Note that for these experiments two (as shown in the diagram) or all three rows of Deiters' cells (usually spanning the distance of 5–10 OHCs) were removed prior to performing the Ca²⁺ imaging experiments.

D, E  Average (D) and maximum (E) Ca²⁺ responses from apical OHCs (P2–P3) induced by photo-damage of non-sensory cells in the GER. Black trace and symbol (wild-type cochlea with Deiters' cells intact) are averages of 85 OHCs from nine recordings; blue trace and symbol (wild-type cochlea in which one or no rows of Deiters' cells were present) are averages of 72 OHCs from nine recordings; red trace and symbol ($Ca_V1.3^{-/-}$ cochlea with Deiters' cells intact but with electrically silent OHCs) are average of 175 OHCs from 10 recordings. Values in (E) are mean ± SEM.

depolarization (Fig 6G; 10 μM ATP: $V_m = -68.9 \pm 3.7$ mV, $n = 4$; 10 μM ATP + 10 μM A-317491: $V_m = -76.6 \pm 2.7$ mV, $n = 4$, $P = 0.0068$, paired *t*-test).

The above data show that Ca²⁺ waves originating in the GER are able to travel to the LER, where the OHCs reside, and induce the release of ATP from the non-sensory Deiters' cells. ATP activates P2X₃ receptors in the basolateral membrane of OHCs, leading to OHC depolarization and an increased open probability of voltage-gated $Ca_V1.3$ Ca²⁺ channels. This depolarization will increase the action potential frequency of the OHCs within the area of the Ca²⁺ wave, thereby increasing the probability of synchronized firing among adjacent OHCs.

**The frequency of large Ca²⁺ waves is reduced in *Cx30⁻/⁻* mice**

In the sensory epithelium of the mammalian cochlea, gap junctions are formed by connexin 26 (Cx26) and Cx30 (Lautermann *et al*,

1998). To test the role of gap junctions in the spread of spontaneous Ca²⁺ activity, we used *Cx30⁻/⁻* mice (Teubner *et al*, 2003) in which the mRNA and protein expression of Cx30 are abolished and those of Cx26 are reduced to only ~ 10% of normal levels during pre-hearing stages (Boulay *et al*, 2013). Despite the loss of connexins, spontaneous and rapid Ca²⁺-dependent signals in developing OHCs were still recorded (Fig 7A). These Ca²⁺ signals occurred in both apical and basal OHCs as shown in wild-type mice (Fig 1), and the number of active OHCs decreased with age (Fig 7B). Electrophysiological recordings from OHCs of *Cx30⁻/⁻* mice showed that their resting membrane potential, ability to fire action potentials and size of the K⁺ currents were not significantly different to those recorded from wild-type cells (Appendix Fig S7). Thus, the intrinsic Ca²⁺ firing activity in developing OHCs was unaffected by the loss of connexins in the non-sensory cells. In agreement with previous findings (Rodriguez *et al*, 2012), the average frequency of Ca²⁺ waves in the GER of *Cx30⁻/⁻* mice (1.3 ± 0.1 events min⁻¹, $n = 36$

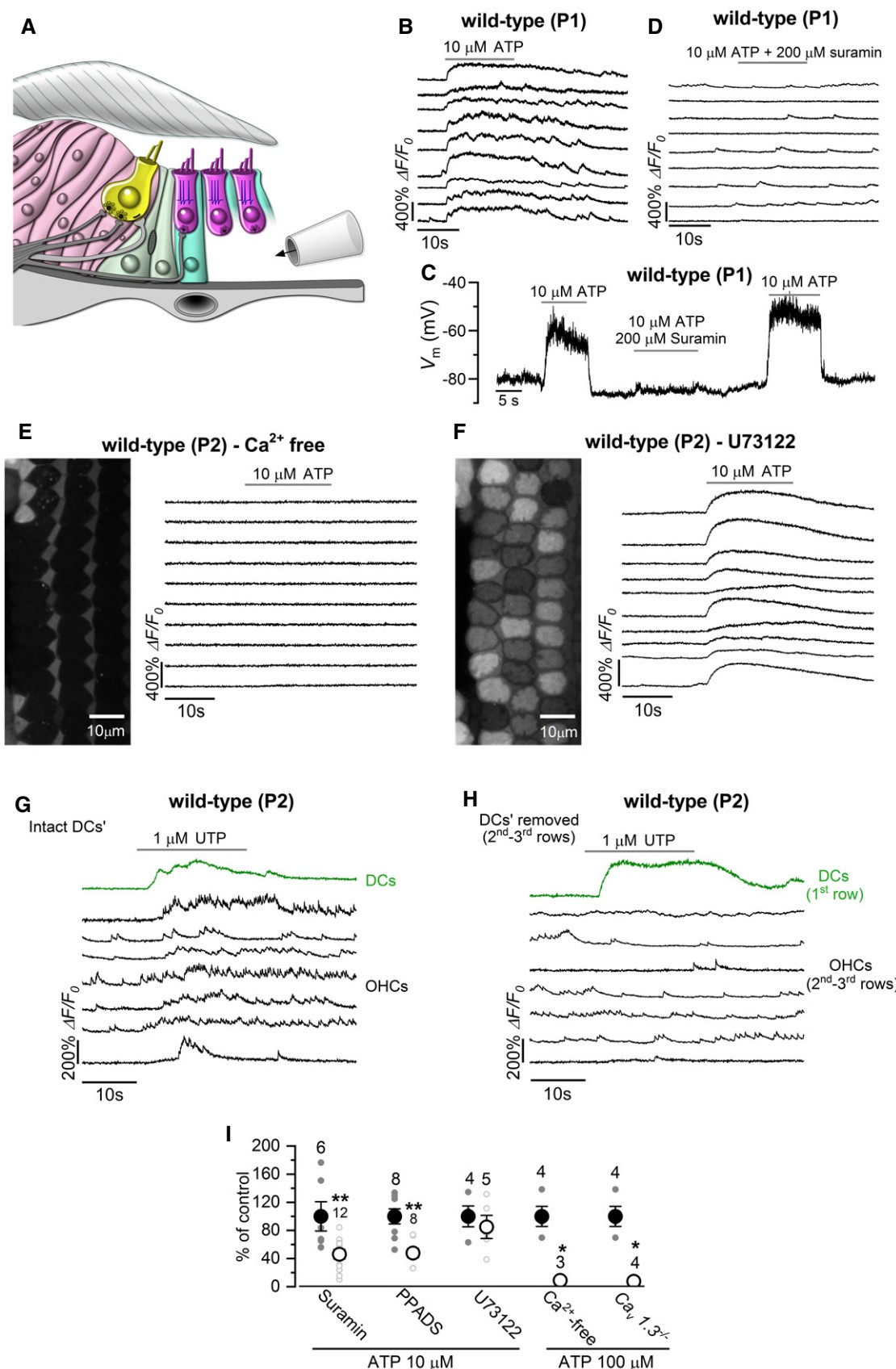

**Figure 5.**

**Figure 5. ATP-induced modulation of the OHC firing activity is mediated by ionotropic purinergic receptors.**

A   Diagram showing a portion of the immature organ of Corti highlighting the experimental approach used to locally perfuse channel blockers and ATP directly to the basolateral membrane of OHCs *in situ*.

B   Representative $\Delta F/F_0$ traces from nine apical OHCs of a P1 wild-type mouse during the application of 10 μM ATP. Data analysis as shown in Fig 1C.

C   Voltage responses in whole-cell current clamp from a P1 OHC of a wild-type mouse during the extracellular application of 10 μM ATP alone or together with 200 μM of the P2X receptor blocker suramin. Suramin largely reduced the ATP-induced OHC depolarization.

D   Representative $\Delta F/F_0$ traces from nine apical OHCs of a P1 wild-type mouse during the application of 10 μM ATP + 200 μM suramin. Note that suramin prevents the occurrence of the large Ca²⁺ signals in OHCs. Data analysis as shown in Fig 1C.

E, F   Representative $\Delta F/F_0$ traces from 10 apical OHCs of P1 wild-type mice (selected from the images in the left panels) during the application of 10 μM ATP in the absence of Ca²⁺ in the extracellular solution (E) or in the presence of the metabotropic P2Y receptor blocker U73122 (10 μM, F). Note that large OHC depolarizations obtained with ATP caused sustained Ca²⁺ signals.

G, H   The P2Y agonist UTP (1 μM) caused increased Ca²⁺ signals in the Deiters' cells, which directly elevate the firing activity of OHCs (G). Note that in the absence of Deiters' cells (H) OHCs did not show Ca²⁺ responses.

I   Histogram showing maximum $\Delta F/F_0$ Ca²⁺ responses in P1–P2 OHCs to the extracellular application of ATP (10 μM or 100 μM) alone (black columns) and together with the purinergic receptor blockers suramin (200 μM) PPADS (100 μM) and U73122 (10 μM), Ca²⁺-free extracellular solution and in $Ca_v1.3^{-/-}$ mice. Experiments were performed in the absence of the Deiters' cells. Responses in each condition are normalized to control experiments, carried out under the same imaging and dye-loading conditions. Number of recordings shown above the columns. *$P < 0.05$, **$P < 0.01$, Mann–Whitney *U*-test. Values are mean ± SEM.

recordings, 11 cochleae, 11 mice, P1–P2) was significantly reduced compared to that of wild-type mice (2.09 ± 0.21 events min⁻¹, $n = 32$ recordings, 7 cochleae, 6 mice, P1–P2, $P = 0.003$, Mann–Whitney *U*-test). Furthermore, the frequency of the larger Ca²⁺ events (> 75 μm: see Fig 2) required for the synchronization of several OHCs was reduced ~ 5-fold in $Cx30^{-/-}$ (0.10 ± 0.03 events min⁻¹, Fig 7C) compared to wild-type mice (0.53 ± 0.11 events min⁻¹, Fig 2, $P < 0.0001$). The few remaining large Ca²⁺ waves in $Cx30^{-/-}$ mice were still able to synchronize the bursting activity of adjacent OHCs (Fig 7C). Similar to wild-type mice (Fig 2G and H), Ca²⁺ signals in OHCs from $Cx30^{-/-}$ mice were significantly stronger for larger Ca²⁺ waves ($P = 0.0106$, *t*-test; Fig 7D). However, the increased Ca²⁺ signals in OHCs were not significantly different between wild-type (Fig 2H) and $Cx30^{-/-}$ (Fig 7D) mice (overall 2-way ANOVA: $P = 0.4526$; Tukey's post-test: < 75 μm: $P = 0.9972$; > 150 μm: $P = 0.8866$). As seen in wild-type mice (Appendix Fig S2A and B), the $r_s^{avg}$ was independent of the amplitude ($\Delta F/F_0$) of the Ca²⁺ signal measured as a pixel average over the entire spread of the Ca²⁺ wave (Appendix Fig S2C and D). Overall, these findings show that the Ca²⁺ signalling from non-sensory cells, although not required for generating spontaneous Ca²⁺ activity in OHCs, is crucial for synchronizing this activity.

## The biophysical characteristics of OHCs from $Cx30^{-/-}$ mice develop normally

In OHCs, the onset of maturation occurs at around P7–P8 when they begin to express a negatively activated K⁺ current $I_{K,n}$ and acquire electromotile activity (Marcotti & Kros, 1999; Abe *et al*, 2007). The expression of the motor protein prestin (Zheng *et al*, 2000; Liberman *et al*, 2002), which drives the somatic motility of OHCs, was normal between wild-type and $Cx30^{-/-}$ mice (see Fig 10). The total K⁺ current in mature OHCs was similar between the two genotypes (Fig 8A–C). The maturation of OHCs is also associated with an increase in cell membrane capacitance (Marcotti & Kros, 1999), which was observed in both genotypes (Fig 8C: $P = 0.4350$). The total outward K⁺ current ($I_K$) and the isolated $I_{K,n}$ recorded from OHCs of $Cx30^{-/-}$ mice (P10–P12) were similar in size to that of wild-type cells ($I_K$: $P = 0.8445$; $I_{K,n}$: $P = 0.2238$, Fig 8C). Mature OHCs are the primary target of the inhibitory olivocochlear efferent fibres that release the neurotransmitter

acetylcholine (ACh; Simmons *et al*, 1996). Efferent inhibition of OHCs by ACh is achieved by Ca²⁺ influx through α9α10-nAChRs activating a hyperpolarizing SK2 current (Oliver *et al*, 2000; Katz *et al*, 2004; Lioudyno *et al*, 2004; Marcotti *et al*, 2004). Mouse OHCs first become highly sensitive to ACh from around the end of the first postnatal week (Katz *et al*, 2004; Marcotti *et al*, 2004), which coincides with their onset of functional maturation (Marcotti & Kros, 1999). In the presence of ACh, depolarizing and hyperpolarizing voltage steps from a holding potential of −84 mV elicited an instantaneous current in wild-type OHCs. This ACh-activated instantaneous current is mainly carried by SK2 channels but also by nAChRs since it is blocked by apamin and strychnine, respectively (Marcotti *et al*, 2004). The ACh-activated current was present in OHCs from wild-type (Fig 8D) and $Cx30^{-/-}$ mice (Fig 8E). The sensitivity of OHCs to ACh was quantified by measuring the steady-state slope conductance at −84 mV of the ACh-sensitive current ($g_{ACh}$), which was obtained by subtracting the control currents from the currents in the presence of 100 μM ACh (Fig 8D and E: see also Marcotti *et al*, 2004). $g_{ACh}$ was similar between wild-type (8.4 ± 1.7 nS, $n = 4$, P12) and $Cx30^{-/-}$ (8.4 ± 1.4 nS, $n = 6$, P10–P12; $P = 0.9843$). We further confirmed that the ACh-induced currents in $Cx30^{-/-}$ OHCs were carried by the nAChRs and SK2 channels since they were blocked by strychnine (at −90 mV: Fig 8F) and a Ca²⁺-free solution (at −40 mV: Fig 8G), respectively, as previously shown in hair cells (Glowatzki & Fuchs, 2000; Oliver *et al*, 2000; Marcotti *et al*, 2004).

## OHC ribbon synapses and afferent fibres are reduced in $Cx30^{-/-}$ and $Ca_v1.3^{-/-}$ mice

The above results demonstrate that pre-hearing $Cx30^{-/-}$ mice, in which OHCs retain their intrinsic Ca²⁺ activity but which have reduced and more spatially confined Ca²⁺ waves in the GER, were able to develop functionally mature OHCs. During the same time window (~ P0–P12), immature IHCs from $Cx30^{-/-}$ mice have been shown to be normal (Johnson *et al*, 2017), indicating that Ca²⁺ waves in the non-sensory cells do not interfere with the normal pre-hearing development of hair cells. Indeed, it has been suggested that the modulation of AP activity in IHCs by the Ca²⁺ waves could be used to refine the afferent auditory pathway (Tritsch *et al*, 2007). Therefore, we made use of the fact that hair cells from $Cx30^{-/-}$ mice

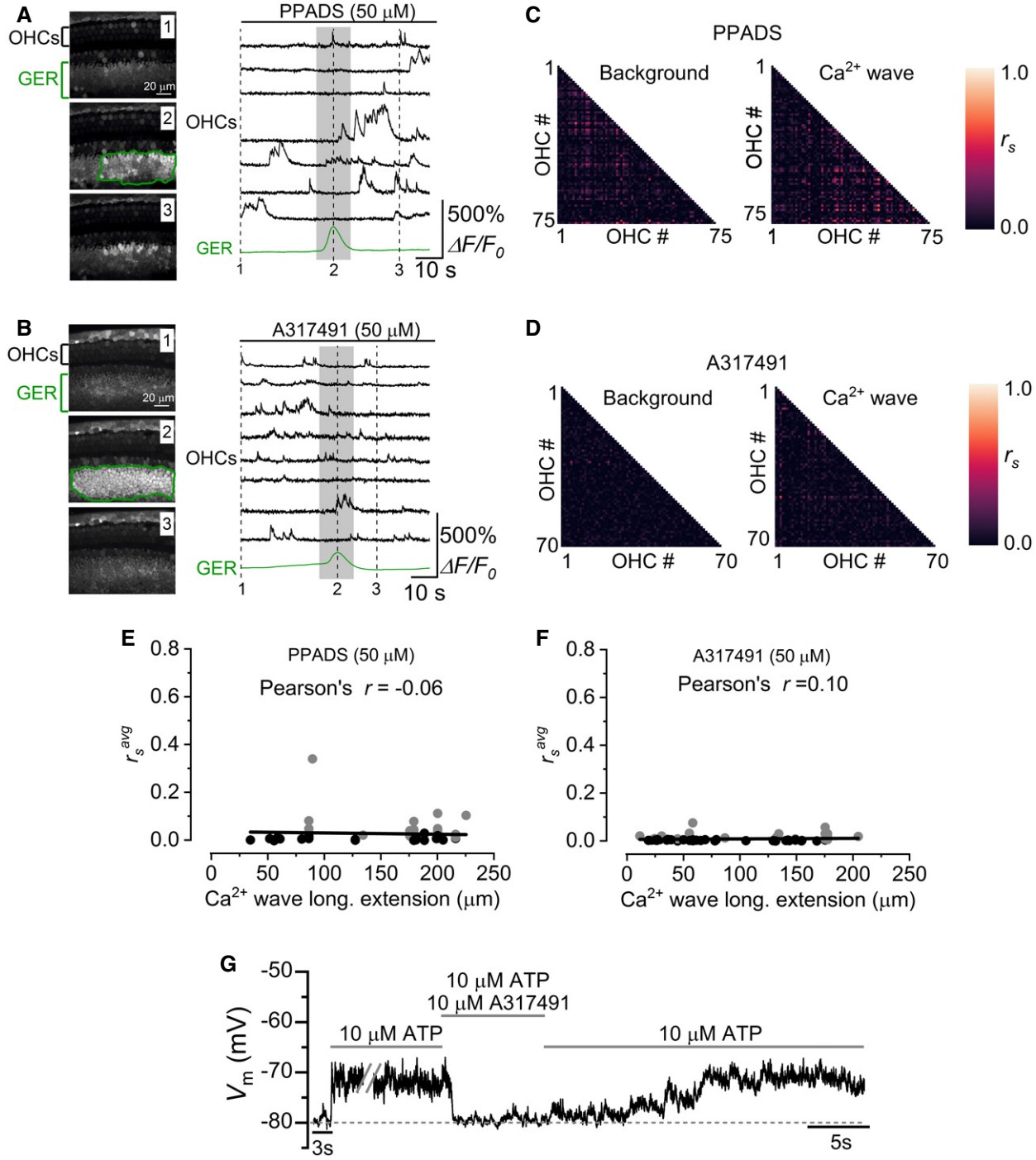

**Figure 6. P2X₃ receptors are implicated in the ATP-induced modulation of the OHC firing activity.**

A, B    Representative $\Delta F/F_0$ traces from apical OHCs of wild-type mice (selected from the images in the left panel) in the continuous presence of the non-selective purinergic antagonist PPADS (A, P2) or the selective P2X₃ antagonist A317491 (B, P1). Note the lack of synchronized Ca²⁺ activity in OHCs despite the presence of a large Ca²⁺ wave in the GER.

C, D    Correlation matrices computed from the Ca²⁺ fluorescence traces of 75 (C: PPADS) and 70 (D: A317491) OHCs from panel (A and B), respectively. See Fig 2D and E legend for more details.

E, F    Average Spearman's rank correlation coefficient ($r_s^{avg}$: see Materials and Methods) between the OHC activity as a function of the longitudinal extension of spontaneous Ca²⁺ waves in the GER from the apical coil of P1–P2 mouse cochleae in the presence of 50 µM PPADS (E, 16 recordings, 11 cochleae, 7 mice) or 50 µM A317491 (F, 17 recordings, 7 cochleae, 6 mice, 1,267 OHCs). Grey and black dots are as described in Fig 2. Solid lines represent a linear fit to the data. Slopes were not significantly different from zero [E: $(-0.05 \pm 0.13)10^{-3}$ µm⁻¹ $P = 0.699$; F: $(0.02 \pm 0.03)10^{-3}$ µm⁻¹ $P = 0.444$].

G    Voltage responses in whole-cell current clamp from a P2 OHC of a wild-type mouse during the extracellular application of 10 µM ATP alone or together with 10 µM of the P2X₃ receptor antagonist A317491. A317491 reversibly blocked the ATP-induced OHC depolarization.

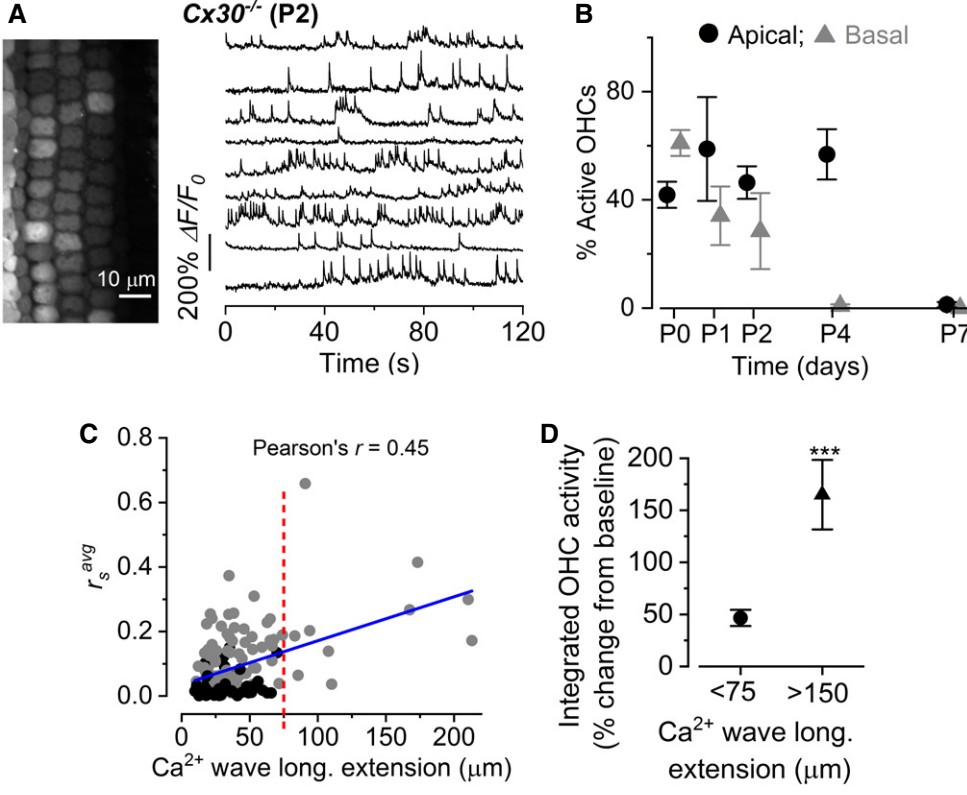

**Figure 7.  Calcium waves from the GER modulate OHC spontaneous Ca²⁺ signalling.**

A  Representative $\Delta F/F_0$ traces from nine apical OHCs of a P2 $Cx30^{-/-}$ mouse (selected from the images in the left panel) recorded at 31°C. Data analysis as shown in Fig 1C.

B  Percentage of apical and basal OHCs showing spontaneous Ca²⁺ signals as a function of postnatal age. Number of total OHCs and recordings from left to right were as follows: apical cochlea 404 and 7; 156 and 3; 368 and 7; 454 and 8; 365 and 8; and basal cochlea 441 and 8; 375 and 6; 328 and 7; 487 and 9; 346 and 6. Values are mean ± SEM.

C  Average Spearman's rank correlation coefficient ($r_s^{avg}$) between the OHC activity as a function of the longitudinal extension (C) of spontaneous Ca²⁺ waves in the GER from the apical coil of P1–P2 mouse cochleae (for additional details, see Fig 2F). Solid line represents a linear fit to the data. The slope was $(1.36 \pm 0.22)10^{-3}$ μm⁻¹, significantly different from zero ($P < 0.0001$; ANOVA, $f$-test).

D  Average increase in the integral of the Ca²⁺ traces in OHCs for small (< 75 μm, 136 recordings) and large (> 150 μm, four recordings) Ca²⁺ waves, as described in wild-type cells (Fig 2H).

are normal during pre-hearing stages to investigate whether the Ca²⁺ signalling in the GER contributes to the refinement of OHC afferent innervation.

The OHC afferent ribbon synapses from wild-type and $Cx30^{-/-}$ mice were investigated before (P4) and after (P10) their onset of functional maturation at ~ P8 (Simmons, 1994). At P4, both wild-type and $Cx30^{-/-}$ OHCs showed a similar number of ribbons ($P = 0.24$, $t$-test: Fig 9A,B and E: CtBP2 puncta in red; Myo7a, blue, was used as the hair cell marker). In mature OHCs (P10), the number of ribbons in $Cx30^{-/-}$ OHCs was about half of that in wild-type OHCs ($P < 0.0001$, $t$-test: Fig 9C,D and E). As a comparison, we also looked at IHCs and found a similar number of ribbons between the two genotypes at both P4 ($P = 0.30$, $t$-test: Appendix Fig S8A,B and E) and P10 ($P = 0.43$, $t$-test: Appendix Fig S8C,D and E), further supporting the evidence that at this age immature IHCs are unaffected by the absence of Ca²⁺ waves (Johnson *et al*, 2017). The requirement for OHC Ca²⁺ signals for the maturation of the afferent synapses was further tested by using knockout mice for $Ca_v1.3$ Ca²⁺ channels ($Cav1.3^{-/-}$), which are required for

hair cell exocytosis. OHCs in P11 $Cav1.3^{-/-}$ mice are present and healthy (Appendix Fig S9). We found a significant reduction in the number of ribbons at P10 ($P < 0.0001$, one-way ANOVA; $P < 0.001$ post-test for wild-type vs. both $Cav1.3^{-/-}$ and $Cx30^{-/-}$ $P > 0.05$ post-test for $Cav1.3^{-/-}$ vs. $Cx30^{-/-}$) but not P4 (one-way ANOVA $P = 0.4497$) compared to wild-type cells (Fig 9E–G).

We then looked at whether the reduction in ribbon synapses was also associated with abnormalities in the afferent fibres innervating mature OHCs. Prestin was used as the OHC marker (Fig 10A–C). At P11 in the apical cochlear region, afferent fibres from spiral ganglion neurons form outer spiral fibres that terminate on the OHCs after long spiral courses (Simmons & Liberman, 1988). Peripherin has been shown to specifically target type II neurons innervating mature OHCs (Hafidi, 1998; Mou *et al*, 1998; Maison *et al*, 2016). We found that type II fibres show peripherin immunoreactivity and course radially from the spiral ganglion to the organ of Corti, cross along the floor of the tunnel of Corti and spiral in a basal direction before giving rise to punctate endings on OHCs (Fig 10D). In the apical region, there were $14.0 \pm 2.0$ (mean ± SD, $n = 3$ mice)

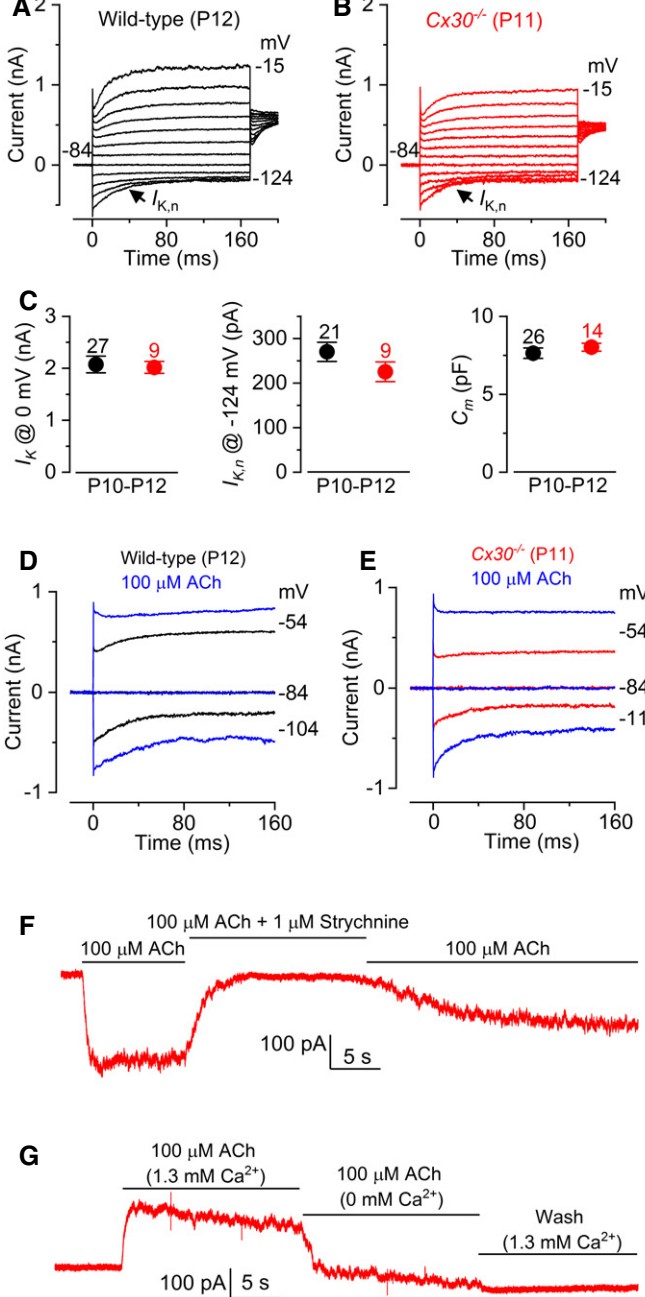

**Figure 8. OHCs from mature *Cx30⁻ᐟ⁻* mice develop normal biophysical properties.**

A, B Current responses in wild-type (A) and *Cx30⁻ᐟ⁻* (B) apical-coil OHCs after their onset of maturity, which occurs at P7–P8. Outward currents were elicited by using depolarizing and hyperpolarizing voltage steps (10 mV increments) from −84 mV to the various test potentials shown by some of the traces.

C Average size of the total outward current measured at 0 mV ($I_K$: left), the isolated $I_{K,n}$ measured as deactivating tail current at −124 mV (middle) and the membrane capacitance ($C_m$: right) of P10–P12 OHCs. Values are mean ± SEM.

D, E Membrane currents recorded from OHCs in wild-type (D, P12) and *Cx30⁻ᐟ⁻* (E, P11) mice before and during superfusion of 100 μM ACh.

F In *Cx30⁻ᐟ⁻* OHCs, the inward current elicited in 100 μM extracellular ACh at −90 mV was reversibly blocked by 1 μM strychnine, indicating the direct involvement of α9α10nAChRs.

G At −40 mV, the outward current in *Cx30⁻ᐟ⁻* OHCs was prevented by an absence of Ca²⁺ in the extracellular solution, indicating the presence of SK2 channels.

basal directions (Fig 10F). The above results provide evidence that the maturation of OHCs ribbon synapses and associated afferents is, at least in part, influenced by the synchronized Ca²⁺ signals in OHCs caused by the Ca²⁺ waves originating in the GER.

## Discussion

We have identified distinct, coordinated Ca²⁺-dependent mechanisms that influence the refinement of OHC innervation. Our evidence shows that the morphological maturation of the afferent synapses and innervation of OHCs requires spontaneous ATP-induced Ca²⁺ waves in the non-sensory cells of the GER, which increase and synchronize the Ca²⁺ activity between several OHCs. Similar Ca²⁺ signalling mechanisms are used to drive the maturation of IHCs (Johnson *et al*, 2013), but they do not appear to be required for the refinement of the pre-synaptic ribbons and postsynaptic afferents. Moreover, in contrast to the ATP-dependent activity modulating IHC action potentials (Wang *et al*, 2015), that influencing OHC Ca²⁺ signals is mediated by ATP-induced activation of P2X₃ receptors, which has distinct functional consequences for OHC maturation and is separated by developmental timing, with OHCs preceding IHCs (Johnson *et al*, 2011, 2017). The data suggest that several distinct patterns of spontaneous, experience-independent Ca²⁺ activity across the auditory sensory epithelium orchestrate the differential maturation of OHCs (afferent innervation) and IHCs (sensory cells Johnson *et al*, 2013) to shape the final stages of auditory organ development.

### OHC activity is synchronized by ATP-induced Ca²⁺ signalling in non-sensory cells

We show that spontaneous intercellular Ca²⁺ signalling activity originating in the non-sensory cells of the greater epithelial ridge (GER) synchronizes Ca²⁺ activity between nearby OHCs via release of ATP from Deiters' cells. This ATP acts directly via P2X receptors on the OHCs. The same activity in the GER synchronizes APs in IHCs (Tritsch *et al*, 2007; Johnson *et al*, 2011; Wang *et al*, 2015; Eckrich *et al*, 2018) but using a different mechanism. In IHCs, ATP indirectly increases the firing activity of IHCs by acting on purinergic

tunnel-crossing outer spiral fibres per 100 μm along the length of the organ of Corti. Compared to wild-type, *Cx30⁻ᐟ⁻* mice had fewer peripherin-labelled outer spiral fibres (Fig 10E), which was matched by a reduction in labelled fibres crossing the tunnel of Corti (7.0 ± 1.0 fibres per 100 μm, *n* = 3 animals) compared to wild-type mice (*P* < 0.01). This result is in agreement with the significantly lower number of ribbon synapses in OHCs from *Cx30⁻ᐟ⁻* mice (Fig 9E). We also found that the cochlea from *Cav1.3⁻ᐟ⁻* mice had a similar reduction in peripherin-labelled fibres (10.0 ± 1.2 per 100 μm distance at 8 kHz, *n* = 3 animals: *P* < 0.05) compared to wild-type mice but not significantly different from *Cx30⁻ᐟ⁻* mice (Fig 10F). Unlike wild-type controls or *Cx30⁻ᐟ⁻* mice (Fig 10D and E), outer spiral fibres in *Cav1.3⁻ᐟ⁻* mice spiralled in both apical and

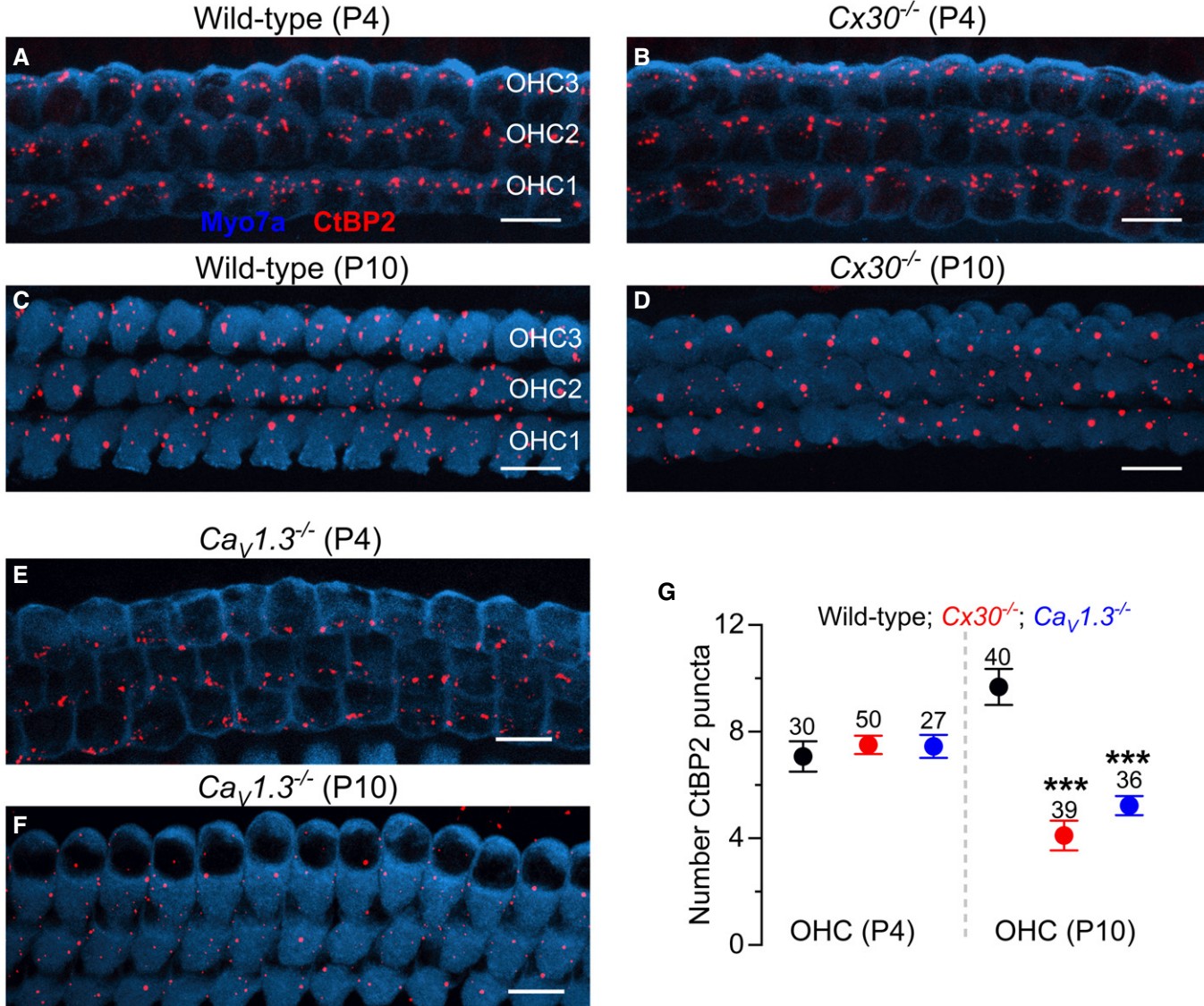

**Figure 9. Ribbon synapses are reduced in *Cx30*$^{-/-}$ and *Ca*$_V$*1.3*$^{-/-}$ OHCs.**

A–D Maximum intensity projections of confocal *z*-stack images that were taken from apical-coil OHCs before (P4) and after (P10) their onset of functional maturation at P8 in wild-type (A and C) and *Cx30*$^{-/-}$ (B and D) mice. Immunostaining for ribbon synapses (CtBP2) is shown in red; Myo7a (blue) was used as the hair cell marker.

E, F Maximum intensity projections as in (A–D) from apical-coil OHCs at P4 (E) and P10 (F) of *Ca*$_V$*1.3*$^{-/-}$ mice.

G Number of ribbons (CtBP2 puncta) in wild-type, *Cx30*$^{-/-}$ and *Ca*$_V$*1.3*$^{-/-}$ OHCs at P4 and P10. Values are mean ± SEM. Number of OHCs analysed is shown above each average data point; four mice were used for each experimental condition. *** indicates $P < 0.001$, one-way ANOVA, Bonferroni post-test. Scale bars 10 μm.

autoreceptors expressed in the non-sensory cells surrounding the IHCs, which leads to the opening of TMEM16A Ca$^{2+}$-activated Cl$^-$ channels and the efflux of K$^+$ in the intercellular space (Wang *et al*, 2015). The expression of these TMEM16A channels seems to follow closely the development of IHCs, and they are absent in the LER (Wang *et al*, 2015). Although P2X$_2$ are the most abundant purinergic receptors in the cochlea, they are mainly expressed in hair cells from the second postnatal week onward throughout adult stages (Järlebark *et al*, 2000), so they are unlikely to mediate the ATP-induced signalling in developing OHCs. P2X$_4$ receptors have been suggested to be present in the developing cochlea based on

pharmacological assays, but these findings have not been confirmed with expression studies (Lahne & Gale, 2010). Of the other known P2X receptors, only P2X$_3$ (Huang *et al*, 2006) and P2X$_7$ (Nikolic *et al*, 2003) have been shown to be transiently expressed during early stages of development. The expression time course of P2X$_3$ receptors seems to match our Ca$^{2+}$-imaging experiments (Fig 1), since a previous study has shown that by P3 they are still present in apical, but no longer in basal OHCs (Huang *et al*, 2006). Indeed, our pharmacological and imaging experiments (Fig 6) demonstrated that P2X$_3$ receptors play a crucial role in mediating the modulation of OHC activity by Ca$^{2+}$ waves originating in the GER. Interestingly, P2X$_3$

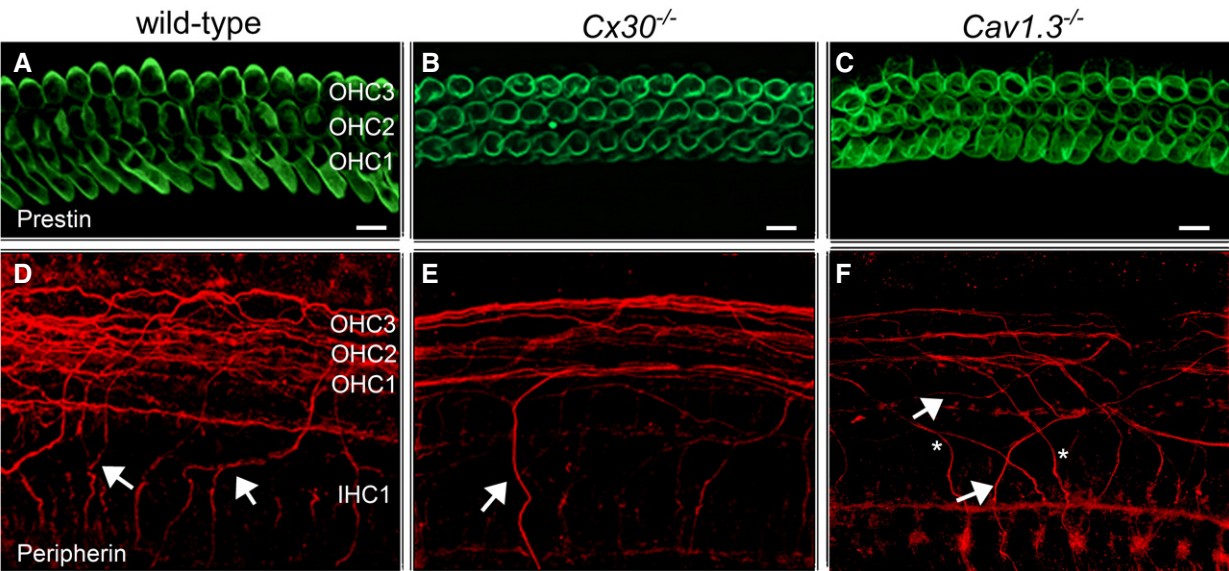

**Figure 10.  Afferent fibres are reduced in $Cx30^{-/-}$ and $Ca_v1.3^{-/-}$ mice.**

Maximum intensity projections of confocal z-stacks taken from the apical cochlear region of wild-type (left column), $Cx30^{-/-}$ (right column) and $Ca_v1.3^{-/-}$ mice at P11 using antibodies against prestin (green) and peripherin (red). Each panel represents a different mouse.

A–C   Prestin labelling was similar between the different mouse strains and as such was used as an OHC marker. Scale bars 10 μm.

D–F   Immunostaining for peripherin (red) highlights outer spiral fibres (arrows) of type II spiral ganglion neurons in the wild-type mouse cochlea (D). These outer spiral fibres cross below IHCs and spiral below OHCs towards the cochlear base. In $Cx30^{-/-}$ (E) and $Ca_v1.3^{-/-}$ (F) mice, there are fewer peripherin-labelled outer spiral fibres than in wild-type. In $Ca_v1.3^{-/-}$ mice (F), the outer spiral fibres travel towards the cochlear base (arrows), as in the wild-type (D), but some also spiral apically (asterisks).

receptors have previously been implicated in early development in the peripheral and central nervous system (Kidd *et al*, 1998).

Although ATP-induced Ca²⁺ signalling from non-sensory cells is crucial for promoting the maturation of IHCs after the onset of hearing (Johnson *et al*, 2017), our data suggest that it does not contribute directly to OHC maturation. However, the time course of maturation of OHCs and IHCs is very different (Knirsch *et al*, 2007; Corns *et al*, 2014). During the time over which OHCs become sensory competent, IHCs are still immature and fire Ca²⁺ action potentials that are thought to drive the functional refinement of the auditory pathway, which mainly occurs during the first week of postnatal development (tonotopic organization in the brainstem: Snyder & Leake, 1997; Kim & Kandler, 2003; spiral ganglion neuron survival: Zhang-Hooks *et al*, 2016; spiral ganglion neuron subtype refinement: Shrestha *et al*, 2018; Sun *et al*, 2018). However, recent results have also indicated that the neuronal diversification process of type I SGNs is already established at birth in mice and as such it is independent of electrical activity (Petitpré *et al*, 2018). IHCs begin to mature towards the end of the second postnatal week (at the onset of hearing), indicating that Ca²⁺ waves from the non-sensory cells are likely to drive different signals to the hair cells during early and later stages of pre-hearing development. There are thus temporally distinct critical windows for the influence of spontaneous Ca²⁺ activity in non-sensory cells on IHCs and OHCs.

## OHC afferent innervation is shaped by ATP-induced intercellular Ca²⁺ signalling in non-sensory cells

In the cochlea, the onset of OHC function is associated with type II spiral ganglion afferent terminals forming extensive arborizations

with several OHCs (Perkins & Morest, 1975; Echteler, 1992). Unlike type I afferent fibres contacting IHCs, type II afferent fibres seem to respond only to the loudest sounds (Robertson, 1984; Brown, 1994), which has led to the assumption that they represent the cochlear nociceptors (Weisz *et al*, 2009; Liu *et al*, 2015). Our imaging experiments show that large spontaneous Ca²⁺ waves originating in the GER (Fig 2F) are able to increase and synchronize bursting activity between OHCs. Considering that the average spiral processes of type II fibres span 215 μm (Weisz *et al*, 2012; Martinez-Monedero *et al*, 2016) and that they contact more than a dozen OHCs (Perkins & Morest, 1975), these Ca²⁺ waves should be sufficient to increase the Ca²⁺ activity of most of the pre-synaptic, immature OHCs that form synapses with each developing afferent fibre. Since OHCs provide an infrequent and weak synaptic input to type II afferent fibres, their suprathreshold excitation would require the summation of the input coming from all OHCs contacting each fibre (Weisz *et al*, 2009, 2012), which in the developing cochlea could be provided by the Ca²⁺ waves originating in the GER. The synchronized activity among nearby OHCs would lead to periodic stimulation of the type II afferent fibres and the activity-dependent refinement of synaptic connections as also seen in the visual system (Katz & Shatz, 1996; Spitzer, 2006). Indeed, we found that the absence of connexins in non-sensory cells, which reduces the frequency and spatial extent of the Ca²⁺ waves and as such OHC synchronization ($Cx30^{-/-}$ mice: Fig 7), leads to a reduced number of ribbon synapses and type II afferent fibres. This finding was also supported by similar results in $Ca_v1.3^{-/-}$ mice, in which OHCs are unable to drive vesicle fusion at their pre-synaptic site. A similar phenotype in the type II afferent innervation was also seen in mice lacking Deiters' cells (Mellado Lagarde *et al*, 2013), corroborating our finding that these non-sensory

cells are crucial for the transfer of information from the GER to the OHCs.

In summary, we propose that in the immature mammalian cochlea, the refinement of the OHC afferent innervation pattern is caused by the increased and synchronized Ca$^{2+}$ activity between neighbouring OHCs, which is provided via Deiters' cells from large Ca$^{2+}$ waves originating in the GER. Overall, our results reveal extraordinary physiological regulation of spontaneous Ca$^{2+}$ signalling in the developing cochlea over discrete and separate time periods, to ensure the correct functional differentiation of neuronal and sensory cells in the maturing auditory system.

# Materials and Methods

### Ethics statement

The majority of the animal studies were performed in the UK and licensed by the Home Office under the Animals (Scientific Procedures) Act 1986 and were approved by the University of Sheffield Ethical Review Committee. Some experiments were performed in the USA, and the animal work was licensed by the Baylor University IACUC (Institutional Animal Care and Use Committee) as established by U.S. Public Health Service.

### Tissue preparation

Apical- and basal-coil OHCs from wild-type mice or transgenic mice of either sex were studied in acutely dissected organs of Corti from postnatal day 0 (P0) to P13, where the day of birth is P0. Transgenic mice include *Cx30$^{-/-}$* (MGI:2447863; Teubner *et al*, 2003) and *Ca$_V$1.3$^{-/-}$* mice (Platzer *et al*, 2000). The genotyping protocols for these transgenic mice were performed as previously described (Platzer *et al*, 2000; Teubner *et al*, 2003). Mice were killed by cervical dislocation, and the organ of Corti dissected in extracellular solution composed of (in mM): 135 NaCl, 5.8 KCl, 1.3 CaCl$_2$, 0.9 MgCl$_2$, 0.7 NaH$_2$PO$_4$, 5.6 D-glucose and 10 Hepes-NaOH. Sodium pyruvate (2 mM), amino acids and vitamins were added from concentrates (Thermo Fisher Scientific, UK). The pH was adjusted to 7.5 (osmolality ~ 308 mmol kg$^{-1}$). The dissected organ of Corti was transferred to a microscope chamber, immobilized using a nylon mesh fixed to a stainless steel ring and viewed using an upright microscope (Olympus BX51 and Nikon FN1, Japan; Leica, DMLFS, Germany; Bergamo II System B232, Thorlabs Inc.). Hair cells were observed with Nomarski differential interference contrast optics (×63 water immersion objective) or Dodt gradient contrast (DGC) optics (×60 water immersion objective) and either ×10 or ×15 eyepieces.

### Single-cell electrophysiology

Membrane currents and voltage responses were investigated either at room temperature (20–24°C) or near body temperature (33–37°C), using Optopatch (Cairn Research Ltd, UK) or Axopatch 200B (Molecular Devices, USA) amplifiers. Patch pipettes, with resistances of 2–3 MΩ, were pulled from soda glass capillaries, and the shank of the electrode was coated with surf wax (Mr Zog's Sex Wax, CA, USA) to reduce the electrode capacitive transient. For whole-cell recordings, the pipette intracellular solution contained

(in mM): 131 KCl, 3 MgCl$_2$, 1 EGTA-KOH, 5 Na$_2$ATP, 5 Hepes-KOH and 10 Na-phosphocreatine (pH was adjusted with 1 M KCl to 7.28; osmolality was 294 mmol kg$^{-1}$). In the experiments designed to investigate the effect of extracellular ATP, Na$_2$ATP was omitted from the above solution. For cell-attached recordings, the pipette contained (in mM): 140 NaCl, 5.8 KCl, 1.3 CaCl$_2$, 0.9 MgCl$_2$, 0.7 NaH$_2$PO$_4$, 5.6 D-glucose and 10 Hepes-NaOH (pH 7.5; 308 mmol kg$^{-1}$). Data acquisition was controlled by pCLAMP software (RRID:SCR_011323) using Digidata 1320A, 1440A or 1550 boards (Molecular Devices, USA). Recordings were low-pass filtered at 2.5 kHz (8-pole Bessel) and sampled at 5 kHz and stored on computer for offline analysis (Origin: OriginLab, USA, RRID: SCR_002815). Membrane potentials in whole-cell recordings were corrected for the residual series resistance $R_s$ after compensation (usually 70–90%) and the liquid junction potential (LJP) of −4 mV measured between electrode and bath solution. The extracellular application of a Ca$^{2+}$-free solution or solutions containing 40 mM KCl, ATP (Tocris Bioscience, UK) or acetylcholine (Sigma-Aldrich, UK) was performed with a multibarrelled pipette positioned close to the patched cells.

### Two-photon confocal Ca$^{2+}$ imaging

For calcium dye loading, acutely dissected preparations were incubated for 40 min at 37°C in DMEM/F12, supplemented with Fluo-4 AM (final concentration 10–20 μM; Thermo Fisher Scientific). The incubation medium contained also pluronic F-127 (0.1%, w/v, Sigma-Aldrich, UK) and sulfinpyrazone (250 μM) to prevent dye sequestration and secretion (Corns *et al*, 2018). Preparations were then transferred to the microscope stage and perfused with extracellular solution for 20 min before imaging to allow for de-esterification.

Ca$^{2+}$ signals were recorded using a two-photon laser-scanning microscope (Bergamo II System B232, Thorlabs Inc., USA) based on a mode-locked laser system operating at 800 nm, 80-MHz pulse repetition rate and < 100-fs pulse width (Mai Tai HP DeepSee, Spectra-Physics, USA). Images were formed by a 60× objective, 1.1 NA (LUMFLN60XW, Olympus, Japan) using a GaAsP PMT (Hamamatsu) coupled with a 525/40 bandpass filter (FF02-525/40-25, Semrock). Images were analysed offline using custom-built software routines written in Python (Python 2.7, Python Software Foundation, RRID:SCR_014795) and ImageJ (NIH) (Schindelin *et al*, 2012). Ca$^{2+}$ signals were measured as relative changes of fluorescence emission intensity ($\Delta F/F_0$). $\Delta F = F - F_0$, where $F$ is fluorescence at time $t$ and $F_0$ is the fluorescence at the onset of the recording.

The extracellular application of solutions containing ATP, ryanodine, the P2X antagonist suramin and PPADS (Tocris), the P2Y agonist UTP (Sigma, UK), the phospholipase C inhibitor U73122 (Tocris Bioscience, UK) and the P2X$_3$ antagonist A-317491 (Sigma) was performed using a Picospritzer or bath application. The pipettes used for local perfusion (diameter 2–4 μm) were pulled from borosilicate glass using a two-step vertical puller (Narishige, Japan). Pressure was kept at a minimum (< 3 psi) to avoid triggering mechanically induced calcium signals. Responses in each experimental condition were normalized to control experiments, carried out on the same day under the same imaging and dye-loading conditions.

Each fluorescence recording consisted of 4,000 frames taken at 30.3 frames per second from a 125 × 125 μm (512 × 512 pixels) region. OHC fluorescence traces were computed as pixel averages

from square ROIs (side = 3.7 μm) centred on each OHC. OHCs were classified as either active or inactive using the following algorithm: (i) imaging traces were smoothed using a moving average temporal filter of length 3. (ii) Slow Ca$^{2+}$ variations and the exponential decay in fluorescence due to photo-bleaching were removed by subtracting a polynomial fit of order 5 to each trace. Detrended traces were normalized to the maximum value in the recording. (iii) The noise floor level was estimated by calculating the power spectral density of the signal using Welch's method and averaging over the large frequencies (greater than 66% of the Nyquist frequency). (iv) A spike inference algorithm [*spikes* (Pnevmatikakis *et al*, 2016); module in the *SIMA* python package (Kaifosh *et al*, 2014)] was used to estimate the (normalized) spike count $s_i$. We then calculated the cumulative spike count $S = \Sigma s_i$ for each trace and considered the cell as active (inactive) if $S$ was above (below) a predetermined threshold. (v) Cells that were classified as active (or inactive) and had a maximum signal below (or above) 4 standard deviations were manually sorted. (vi) The entire dataset was independently reviewed by two experimenters. Cells that had discording classification based on the above criteria (69 out of 2,229 at body temperature and 30 out of 5,217 at room temperature) were removed from the analysis. For the experiments in which we calculated the Ca$^{2+}$ spike frequency from Ca$^{2+}$ imaging data (Appendix Fig S1E), we first estimated the number of spikes from the posterior marginal distribution of 1,000 samples of spike trains produced by the Markov chain Monte Carlo (MCMC) spike inference algorithm described in Pnevmatikakis *et al* (2016). The average frequency was then computed by dividing the number of spikes by the total duration of the recording (133 s).

For recording spontaneous activity in the GER, we increased the field of view to a 182 × 182 μm region, which was dictated by the ability to detect the full extension of a Ca$^{2+}$ wave in the GER and to maintain a sufficient spatial resolution to resolve the activity of individual OHCs with good signal-to-noise ratio. Under these conditions, the average length of apical coil used for these experiments was 188 ± 4 μm, since some preparations were positioned diagonally in the field of view. Under this recording condition, some large Ca$^{2+}$ waves were underestimated because they travelled beyond the field of view. Time-series images were corrected for motion using a rigid-body spatial transformation, which does not distort the image (spm12; www.fil.ion.ucl.ac.uk/spm). Recordings showing large drifts of the preparation were discarded from the analysis to avoid potential artefacts in the computation of correlation. Calcium waves were manually identified using thresholding, and a ROI was drawn around the maximum extension of each multicellular calcium event. Only events that initiated within the field of view of the microscope were considered for this analysis. GER fluorescence traces were computed as ROI pixel averages, and as such they give an indication of the average cytosolic calcium increase in non-sensory cells participating in the propagation of the Ca$^{2+}$ wave. To measure the degree of correlation between OHCs during Ca$^{2+}$ activity in the GER, we first computed the pairwise Spearman's rank correlation coefficient ($r_s$) between every pair of OHCs in the field of view (Fig 2D and E). We then averaged $r_s$ using Fisher's $z$-transformation

$$z = \mathrm{arctanh}(r_s)$$

$$r_s^{avg} = \tanh(\langle z \rangle)$$

We take $r_s^{avg}$ as a measure of the average degree of coordination of the activity of neighbouring OHCs.

To test for the increase in coordinated OHC activity, we used the Mann–Whitney $U$-test (one sided) to check whether OHC correlation coefficients during spontaneous Ca$^{2+}$ activity in the GER were significantly ($P < 0.001$) greater than those computed over a time window of 13.2 s (400 frames) during which no Ca$^{2+}$ waves were observed in the GER.

To quantify the change in OHC activity during the Ca$^{2+}$ waves in non-sensory cells, we measured the integral of the Ca$^{2+}$ trace in the same 400 frames (see above) in the absence of Ca$^{2+}$ waves (baseline) and during Ca$^{2+}$ waves. Traces were smoothed using the Savitzky–Golay filter (window length = 11, polynomial order = 1) and normalized to the baseline $F_0$ before computing the integral.

Photo-damage-induced Ca$^{2+}$ waves were triggered by applying high-intensity laser pulses using a second mode-locked laser system operating at 716 nm (Mai Tai HP, Spectra-Physics, USA). The laser was merged into the excitation light path using a long-pass dichroic mirror (FF735-Di02, Semrock) and focused on the preparation through the imaging objective (LUMFLN60XW, Olympus, Japan). Two galvanometric mirrors were used to steer the laser beam across the photo-damage area (6.6 × 8.4 μm), which typically comprised one or two non-sensory cells of the GER. The number of repetitions, and thus the total amount of energy delivered, was set to the minimum able to trigger a Ca$^{2+}$ wave (typically five repetitions, lasting 165 ms in total).

## Immunofluorescence microscopy

Dissected inner ears from wild-type and $Cx30^{-/-}$ and $Ca_V1.3^{-/-}$ mice ($n \geq 3$ for each set of experiment) were fixed with 4% paraformaldehyde in phosphate-buffered saline (PBS, pH 7.4) for 5–20 min at room temperature. Cochleae were microdissected, rinsed three times for 10 min in PBS and incubated for 1 h at room temperature in PBS supplemented with 5% normal goat or horse serum and 0.3% Triton X-100. The samples were then incubated overnight at 37°C with the primary antibody in PBS supplemented with 1% of the specific serum. Primary antibodies were as follows: mouse anti-myosin7a (1:1,000, DSHB, #138-1), rabbit anti-myosin7a (1:200, Proteus Biosciences, #25-6790), rabbit anti-peripherin (#AB1530, 1:200, Millipore), mouse anti-CtBP2 (1:200, Biosciences, #612044) and rabbit anti-prestin (1:2,000, kindly provided by Robert Fettiplace). All primary antibodies were labelled with species appropriate Alexa Fluor secondary antibodies for 1 h at 37°C. Samples were then mounted in VECTASHIELD. The $z$-stack images were captured either with a LSM 800 with Airyscan (Carl Zeiss) system with GaAsP detectors or with a Nikon A1 confocal microscope. Image stacks were processed with Fiji Image Analysis software.

## Statistical analysis

Statistical comparisons of means were made by Student's two-tailed $t$-test or, for multiple comparisons, analysis of variance (one-way or 2-way ANOVA followed by Bonferroni's test) and Mann–Whitney $U$-test (when normal distribution could not be assumed) were applied. $P < 0.05$ was selected as the criterion for statistical significance. Mean values are quoted in text and figures as means ± SEM (electrophysiology and imaging) and ±SD (fibre counting). Only mean

values with a similar variance between groups were compared. Animals of either sex were randomly assigned to the different experimental groups. No statistical methods were used to define sample size, which was defined based on previous published similar work from our laboratory. Animals were taken from multiple cages and breeding pairs. The electrophysiological and morphological (but not imaging) experiments were performed blind to animal genotyping.

**Expanded View** for this article is available online.

## Acknowledgements

The authors thank Joerg Striessnig (University of Innsbruck) for providing the Ca$_V$1.3$^{-/-}$ mice; Aubrey Hornak, Andrew Cox and Jemima McCluskey (Baylor University) for their technical assistance with the immunostaining experiments; Michelle Bird (University of Sheffield) for her assistance with the transgenic mouse colonies; and Maria Pakendorf (University of Sheffield) for helping with the genotyping. This work was supported by the Wellcome Trust to W.M. (102892/Z/13/Z), the National Institute on Deafness and Other Communication Disorders to D.D.S. (K18 DC013304), a 2015–2016 Fulbright Scholar Award to D.D.S and DFG CRC 894 and CRC 1027 to J.E. C.J.K. was supported by the MRC (MR/K005561/1).

## Author contributions

FC, AH, J-YJ, SLJ, FS, JE, JO, DDS and WM collected and analysed the data. All authors helped with the interpretation of the results. FC, AH, MCH, CJK, DDS and WM wrote the paper. FM helped with the design of the 2-photon imaging system. WM conceived and coordinated the study. SLJ is a Royal Society University Research Fellow.

## Conflict of interest

The authors declare that they have no conflict of interest.

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
