## [Review Process File · The EMBO Journal]

Coordinated calcium signaling activity of cochlear sensory and non-sensory cells refines afferent innervation of outer hair cells

Federico Ceriani, Aenea Hendry, Jing-Yi Jeng, Stuart L. Johnson, Friederike Stephani, Jennifer Olt, Matthew C. Holley, Fabio Mammano, Jutta Engel, Corné J. Kros, Dwayne D. Simmons, Walter Marcotti

Review timeline:	Submission date:	16th May 2018
	Editorial Decision:	9th Jul 2018
	Revision received:	17th Oct 2018
	Editorial Decision:	23rd Nov 2018
	Revision received:	11th Dec 2018
	Accepted:	18th Jan 2019

Editor: Elisabetta Argenzio

Transaction Report:

1st Editorial Decision

9th Jul 2018

Thank you for submitting your manuscript on a role for calcium waves in cochlear non-sensory cells during the development of the auditory afferent innervation to The EMBO Journal. We have now received two referee reports on your study, which are enclosed below for your information.

As you can see, while referee #1 is overall more positive, referee #2 raises major issues that need to be addressed before s/he can support publication at The EMBO Journal. In particular, this referee finds that your claims on direct causalities between reduced calcium wave activity and altered afferent outer hair cell innervation are not sufficiently supported by the current data. Thus, this referee requests additional experimental evidence in support of the model proposed. Also, s/he points out that OHC coordination and the purinergic signaling from Deiters cells to OHCs during calcium waves need deeper investigation.

Given the overall interest of your study, I would thus like to invite you to revise the manuscript in response to the referee reports. Addressing these issues through decisive additional data as suggested would be essential to warrant publication in The EMBO Journal. Please note that it is The EMBO Journal's policy to allow only one single major round of revision and will therefore be critical to resolve the main concerns at this stage.

REFEREE REPORTS:

Referee #1:

The manuscript from Ceriani et al. provides insight into the development of afferent innervation of cochlear OHCs. Previous work has focused on IHCs, and several groups have shown an interesting pathway that involves Ca²⁺ waves, ATP release from Deiters' cells, and Ca²⁺ signaling in the complex of IHCs and their surrounding supporting cells. The present submission shows that a related but distinct pathway controls OHC activity and innervation, with Ca²⁺ waves in the greater

epithelial ridge leading to ATP release which directly activates OHCs via P2X receptors. Because the Ca²⁺ waves are widespread, nearby OHCs are activated in a coordinated fashion, which leads to coordinated synaptic release and, eventually, synapse reinforcement and afferent fiber connectivity. This OHC activity occurs in a narrow developmental window, earlier than the corresponding window for IHCs. The data strongly support the authors' hypothesis, and the manuscript is convincing. The videos supporting the data in the manuscript are very clear and nicely illustrate the phenomena they are characterizing.

I have no major comments, and only a few minor comments:

Line 105: comma does not belong.

Line 143-144: usually when DCs are removed and OHCs are patched, the OHC is a little ways away from the area of damage, isn't it? Could the OHC contacting a DC be damaged if the DC is removed? How can you show that there is not a problem with the proximal OHC?

Line 142-147: very long sentence with two "and"s.

Line 256: comma does not belong.

Line 279: a concluding sentence that summarizes this section would be nice.

Line 288: it is unclear which Ca²⁺ activity "this" refers to. The sentence structure would argue that it refers to Ca²⁺ signaling influencing IHCs, but the context suggests OHCs. Recommend rewriting this sentence to clarify.

Line 294: why the colon?

Referee #2:

This manuscript hooks up with a number of recent papers that address the occurrence and mechanisms of intrinsic activity in the auditory periphery and its role in shaping the auditory pathway during pre-sensory development.

Here, Ceriani et al. identify spontaneous spiking activity and Ca²⁺ dynamics in outer hair cells during postnatal development and demonstrate that this activity is modulated by spontaneously occurring Ca²⁺ waves originating within and traveling across non-sensory cells in the greater epithelial ridge. They further go on to show that the signal transmission from the GER to OHCs is lost when Deiter's cells are mechanically removed and show that OHC cell activity can be increased by purinergic signaling involving P2X receptors, consistent with the idea that transmission of the Ca²⁺ wave to OHCs occurs via ATP release from Deiter's cells. Finally, they use a genetic model with reduced Ca²⁺ wave frequency (connexin 30 knockout mice) to address potential consequences of the GER-to-OHC-signaling. While they find that functional maturation of OHCs appears unchanged, synaptic connectivity to type II afferents is altered in the Cx26 model, such that both the number of presynaptic ribbons in OHCs and the number of afferent fibers are reduced. They combine all of these observations to suggest a model in which non-sensory Ca²⁺ waves synchronize OHC activity (via Deiter's cells) and this synchronized OHC activity is essential for refinement or maintenance of their afferent (type II) innervation pattern during postnatal development.

Overall, the data include many new and highly interesting observations, including electrical activity of developing OHCs, modulation by non-sensory Ca²⁺ waves, and by purinergic ionotropic signaling.

Also, the proposed function in shaping afferent innervation of OHCs is very attractive, given the idea that type II afferents, which signal high level (noxious) sound, probably also need synchronous synaptic inputs from a population of OHCs for excitation in the mature cochlea.

However, the experimental evidence shows a number of gaps with respect to the full model proposed. Maybe most importantly, the main conclusion, namely that coordinated (synchronized) activity of OHCs is required for refining type II afferents, lacks stringent experimental support. The

proposal mainly relies on the Cx30 ko that has reduced Ca²⁺ wave activity on the one hand and altered afferent OHC innervation on the other hand. This only provides quite indirect evidence for a causative relation between both. Moreover a connection between both knockout phenotypes via modulation of hair cell activity is not shown (even though this explanation appears attractive). Actually, there is even no experiment that shows that hair cell electrical/Ca²⁺ activity is required for securing the correct innervation pattern at all. Yet, it seems that appropriate experimental models to address this question should be available. E.g., what about type II innervation pattern in the Cav1.3 ko that lacks OHC activity entirely (Fig. 1)? Other helpful models may include release-deficient mice (otoferlin ko, etc...).

Another uncertainty relates to the suggestion of OHC coordination. While the authors stress increased cross-correlation of activity between individual hair cells, it seems that the predominant effect of the Ca²⁺ waves is to increase OHC activity (which is in fact quantified only for one specific experiment, Fig. 5D). So the more economical explanation might be that an increase in OHC activity drives the observed development of innervation. A third open end is the question of purinergic signaling from Deiters cells to OHCs during Ca²⁺ waves. Removal of the cells provides some evidence, but could also damage other mechanisms. The identification of involvement of OHC P2X receptors should allow for pharmacological interruption of GER-to-OHC coupling to clarify this issue.

Specific points:

1. Synchronization/coordination of OHC activity. What is really measured by the cross-correlation coefficient? Visual inspection of the traces, e.g. Fig3C, does not readily reveal common patterns in the fine structure of the OHC activity traces, i.e. temporal synchronization of individual spikes. Is it possible that apparent correlation simply results from overall increase in activity, making all traces more similar (less amplitude divergence between OHCs)? It seems that an average of all OHC traces would result in a wave very similar to the averaged Ca²⁺ wave in the GER.
2. It is further not entirely clear which pairs of hair cells were examined for correlation. Are pairs only immediately neighbouring couples or any pair of OHCs even if separated by other (rows of) cells?
3. Why is activity level never analyzed separately from correlation, except for Fig. 5D,E.?
4. It is important for the mechanistic conclusions that spontaneous activity is not affected in the Cx30 mice. However, a quantitative comparison is missing.
5. Experiments with Deiters cells removed were done differently from the rest of the experiments, i.e. induction of the Ca²⁺ event by photodamage versus spontaneous Ca²⁺ waves. Why? Do both approaches examine the same process? At least the hair cells close to the damage site appear to exhibit quite distinct Ca²⁺ signals, i.e. slow and graded Ca²⁺ elevation rather than spikes (Fig. 5B).
6. What is the quantitative effect of Deiters cell removal on spontaneous OHC spiking (only anecdotic data shown, Suppl. fig. 2A)?
7. Quantification/statistics are missing for Data shown in Fig. 2 (Ca²⁺-free, Cav1.3. ko, ryanodine)
8. Data shown in Fig. 4 lack statistics and demonstration of reproducibility (how many observations, etc.)

Ceriani et al. 'Coordinated Ca^{2+} activity of cochlear sensory and non-sensory cells refines OHC afferent innervation' – submission of the revised version of manuscript EMBOJ-2018-99839.

Summary of major changes in response to the Reviewers Comments

We thank the Reviewers for their numerous constructive comments, which have helped to strengthen the MS. In order to address these comments we have performed several new experiments, which have resulted in 6 new Figures (Figure 6; Supplementary Figure 3, 4, 6, 7,8) and the addition of new data/analysis to another 4 Figures (Figure 2, 7, 10; Supplementary Figure 1).

Here are some of the major changes in the revised version of the MS:

- 1) In response to **Reviewer 1 & 2** we have performed additional experiments to test whether the removal of Deiters' cells affects the biophysical characteristics of postnatal OHCs (see below). Our new data show that the resting membrane potential, ability to fire action potentials and size and shape of the K^+ current of OHCs were not significantly affected by the removal of the DCs (new **Supplementary Fig 3A-G**).
- 2) We have also performed several additional experiments to provide further evidence for the presence of purinergic signalling coupling the Deiters' cells and OHCs during calcium waves (**Editorial Remark #2** and **Reviewer 2**). Using the intact cochlear preparation and 2-photon imaging, we found that the non-selective purinergic antagonist PPADS largely prevents the modulation of OHC calcium spike activity by the calcium waves propagating among non-sensory cells of the GER and LER (new **Fig 6A,C,E**). Using both 2-photon imaging (new **Fig 6B,D,F**) and electrophysiology (new **Fig 6G**), we have also discovered that P2X3 purinergic receptors are essential for this functional coupling.
- 3) We have also added new immunolabelling experiments (revised **Fig 10**; new **Supplementary Fig 7, 8**) further supporting the claim that the correct afferent innervation (and ribbon synapses) is modulated by OHC activity (**Editorial Remark #1** and **Reviewer 2**). For these experiments, we used $\text{Ca}_v1.3$ KO mice, as suggested by **Reviewer 2**; $\text{Ca}_v1.3$ calcium channels are required for exocytosis in both OHCs and IHCs. $\text{Ca}_v1.3$ KO mice were preferred to the other model proposed by **Reviewer 2** (otoferlin KO mice) because it is well established that exocytosis in cochlear hair cells is, at least initially (from late embryonic stages to ~P2), otoferlin-independent. This is the time when the calcium waves from the GER are more active in influencing OHC activity. To further verify our additional experiments on $\text{Ca}_v1.3$ KO mice, we have asked Prof Jutta Engel (Germany), who is an expert in calcium channel expression in the cochlea, to help with this additional work; as such Prof Engel and her PhD student have been included in the author list.

Editor Remarks: this referee finds that your claims on direct causalities between reduced calcium wave activity and altered afferent outer hear cell innervation are not sufficiently supported by the current data. Thus, this referee requests additional experimental evidence in support of the model proposed. Also, s/he points out that OHC coordination and the purinergic signaling from Deiters cells to OHCs during calcium waves need deeper investigation.

As mentioned above, and also discussed in greater detail in the specific replies to Reviewer 2 (see below), we have addressed both points (**Editor Remarks**), and additional crucial remarks made by both Reviewers, by performing a substantial number of new experiments.

Referee #1:

The manuscript from Ceriani et al. provides insight into the development of afferent innervation of cochlear OHCs. Previous work has focused on IHCs, and several groups have shown an interesting pathway that involves Ca²⁺ waves, ATP release from Deiters' cells, and Ca²⁺ signaling in the complex of IHCs and their surrounding supporting cells. The present submission shows that a related but distinct pathway controls OHC activity and innervation, with Ca²⁺ waves in the greater epithelial ridge leading to ATP release which directly activates OHCs via P2X receptors. Because the Ca²⁺ waves are widespread, nearby OHCs are activated in a coordinated fashion, which leads to coordinated synaptic release and, eventually, synapse reinforcement and afferent fiber connectivity. This OHC activity occurs in a narrow developmental window, earlier than the corresponding window for IHCs. The data strongly support the authors' hypothesis, and the manuscript is convincing. The videos supporting the data in the manuscript are very clear and nicely illustrate the phenomena they are characterizing.

I have no major comments, and only a few minor comments:

Line 105: comma does not belong.

Done

Line 143-144: usually when DCs are removed and OHCs are patched, the OHC is a little ways away from the area of damage, isn't it? Could the OHC contacting a DC be damaged if the DC is removed? How can you show that there is not a problem with the proximal OHC?

This is a very good question, which also applies to most of the classical literature investigating the biophysical properties of isolated (e.g. Mammano & Ashmore, 1996) or *in situ* (Marcotti & Kros, 1999) OHCs. In the latter DCs are normally removed in order to gain access to the OHC basolateral membrane. In order to understand whether the biophysical properties of OHCs change in the absence of DCs, we performed additional electrophysiological recordings since they allow a more detailed investigation of OHC function. Recordings were done from OHCs in the absence or presence of all three rows of DCs (new **Supplementary Fig 3A-G**); in the latter, the patch pipette was advanced through the tissue with minimal positive pressure in order to minimize any disturbance. The results show that all parameters investigated (size and shape of the K⁺ current, ability to fire action potentials and resting membrane potential) were not significantly different between the two recording conditions. This is now added into the revised text (**ln. 158-162**).

Line 142-147: very long sentence with two "and"s.

The sentence has been changed.

Line 256: comma does not belong.

Done

Line 279: a concluding sentence that summarizes this section would be nice.

The sentence has been added (**ln. 335-338**)

Line 288: it is unclear which Ca²⁺ activity "this" refers to. The sentence structure would argue that it refers to Ca²⁺ signaling influencing IHCs, but the context suggests OHCs. Recommend rewriting this sentence to clarify.

The sentence has been changed (ln. 346-352)

Line 294: why the colon?

Removed

Referee #2:

This manuscript hooks up with a number of recent papers that address the occurrence and mechanisms of intrinsic activity in the auditory periphery and its role in shaping the auditory pathway during pre-sensory development.

Here, Ceriani et al. identify spontaneous spiking activity and Ca²⁺ dynamics in outer hair cells during postnatal development and demonstrate that this activity is modulated by spontaneously occurring Ca²⁺ waves originating within and traveling across non-sensory cells in the greater epithelial ridge. They further go on to show that the signal transmission from the GER to OHCs is lost when Deiter's cells are mechanically removed and show that OHC cell activity can be increased by purinergic signaling involving P2X receptors, consistent with the idea that transmission of the Ca²⁺ wave to OHCs occurs via ATP release from Deiter's cells. Finally, they use a genetic model with reduced Ca²⁺ wave frequency (connexin 30 knockout mice) to address potential consequences of the GER-to-OHC-signaling. While they find that functional maturation of OHCs appears unchanged, synaptic connectivity to type II afferents is altered in the Cx26 model, such that both the number of presynaptic ribbons in OHCs and the number of afferent fibers are reduced.

They combine all of these observations to suggest a model in which non-sensory Ca²⁺ waves synchronize OHC activity (via Deiter's cells) and this synchronized OHC activity is essential for refinement or maintenance of their afferent (type II) innervation pattern during postnatal development.

Overall, the data include many new and highly interesting observations, including electrical activity of developing OHCs, modulation by non-sensory Ca²⁺ waves, and by purinergic ionotropic signaling.

Also, the proposed function in shaping afferent innervation of OHCs is very attractive, given the idea that type II afferents, which signal high level (noxious) sound, probably also need synchronous synaptic inputs from a population of OHCs for excitation in the mature cochlea.

However, the experimental evidence shows a number of gaps with respect to the full model proposed. Maybe most importantly, the main conclusion, namely that coordinated (synchronized) activity of OHCs is required for refining type II afferents, lacks stringent experimental support. The proposal mainly relies on the Cx30 ko that has reduced Ca²⁺ wave activity on the one hand and altered afferent OHC innervation on the other hand. This only provides quite indirect evidence for a causative relation between both. Moreover a connection between both knockout phenotypes via modulation of hair cell activity is not shown (even though this explanation appears attractive). Actually, there is even no experiment that shows that hair cell electrical/Ca²⁺ activity is required for securing the correct innervation pattern at all. Yet, it seems that appropriate experimental models to address this question should be available. E.g., what about type II innervation pattern in the Cav1.3 ko that lacks OHC activity entirely (Fig. 1)? Other helpful models may include release-deficient mice (otoferlin ko, etc...).

As indicated by this reviewer, we have now added new evidence indicating that calcium-dependent activity in OHCs is required for refining afferent innervation. For these experiments (see also general statement on page 1, point 3), we used Cav1.3 KO mice, which were preferred to the

other model proposed (otoferlin KO mice) since it is known that exocytosis is initially otoferlin-independent in cochlear hair cells (from late embryonic stages to ~P2). This is the time when the calcium waves from the GER are more active in influencing OHC activity. These new data are shown in the revised **Fig 10** (text: **ln. 324-338**).

Another uncertainty relates to the suggestion of OHC coordination. While the authors stress increased cross-correlation of activity between individual hair cells, it seems that the predominant effect of the Ca²⁺ waves is to increase OHC activity (which is in fact quantified only for one specific experiment, Fig. 5D). So the more economical explanation might be that an increase in OHC activity drives the observed development of innervation.

The reviewer is correct in his/her assumptions. Our data show that calcium waves increase the frequency of calcium spikes in OHCs, which will also increase the probability of synchronized firing (**ln. 121-135**).

While an increase in firing frequency of one single OHC may help strengthen its synaptic contacts with type II fibres, it would not be sufficient to trigger firing in the type II neuron, due to its relatively high threshold (about 24 mV positive to their resting membrane potential: Weisz et al. 2014). This will require the input of several OHCs, since an estimate of at least 6 EPSPs have to occur simultaneously (or with a maximal delay of 7 ms, estimated from mathematical modelling: Weisz et al. 2014) for the fibre to reach the threshold. As mentioned above, by increasing the firing frequency of all OHCs, calcium waves will increase the probability that a postsynaptic type II neuron is brought to the firing threshold. Moreover, our data from Connexin 30 mice, which have reduced frequency of calcium waves but normal intrinsic OHC activity and fail to acquire the normal pre-and post-synaptic afferent wiring, suggests that the coordinated activity of OHCs is crucial for this developmental aspect.

In the revised MS we have also provided more quantification of the data presented (see our replies to several specific points below).

A third open end is the question of purinergic signaling from Deiters cells to OHCs during Ca²⁺ waves. Removal of the cells provides some evidence, but could also damage other mechanisms. The identification of involvement of OHC P2X receptors should allow for pharmacological interruption of GER-to-OHC coupling to clarify this issue.

We are particularly grateful for this point because it has allowed us to further extend the investigation into the nature of the receptors involved in the coupling between Deiters' cells and OHCs. Using the intact epithelium we found that 50 μ M of the non-selective purinergic receptor antagonist PPADS was able to uncouple calcium waves from OHCs; large calcium waves were no longer able to influence OHC activity and as such prevented the previously observed co-ordinated firing among OHCs (see new **Fig 6A,C,E**). Moreover, we performed additional imaging and electrophysiological experiments and discovered that P2X3 purinergic receptors are required for the calcium waves to influence OHC calcium spike activity (new **Fig 6B,D,F,G**) (text: **ln. 209-230**).

Specific points:

1. Synchronization/coordination of OHC activity. What is really measured by the cross-correlation coefficient? Visual inspection of the traces, e.g. Fig3C, does not readily reveal common patterns in the fine structure of the OHC activity traces, i.e. temporal synchronization of individual spikes. Is it possible that apparent correlation simply results from overall increase in activity, making all traces

more similar (less amplitude divergence between OHCs)? It seems that an average of all OHC traces would result in a wave very similar to the averaged Ca²⁺ wave in the GER.

As mentioned above (see also **ln. 121-135**), calcium waves from non-sensory cells cause an increase in calcium spike frequency in OHCs. Due to the relative long fluorescence decay time constant of OHC calcium signals, (the fluorescence signal generated by a single Ca²⁺ spike has a time constant of ~300 ms, **Ceriani et al, 2016**), which effectively imposes a low-pass filter on the spike train, high frequency Ca²⁺ bursts appear as graded increases in fluorescence (or as smaller spikes superimposed on a graded increase in fluorescence). This, and the fact that we are computing our correlation coefficients in a time window of 13.2 s, means that with our correlation analysis we measure correlation of bursts, rather than single spikes, in different OHCs. In other words, the average correlation coefficient measures how “strongly” OHCs in our recording region are co-active in a given time interval.

2. It is further not entirely clear which pairs of hair cells were examined for correlation. Are pairs only immediately neighbouring couples or any pair of OHCs even if separated by other (rows of) cells?

Each OHC is examined in respect to any other OHC in the field of view. To further stress this point, we included a visual representation of the correlation matrices for the representative experiments shown in **Fig 2** (panels D,E) and **Fig 6**.

3. Why is activity level never analyzed separately from correlation, except for Fig. 5D,E.?

Calcium wave level is now shown for wild-type OHCs (**Fig 2G**), Cx30 KO mice (**Fig 7D**), ATP experiments (**Fig 5I**) and 0-Ca²⁺ (**Supplementary Fig 1C**).

4. It is important for the mechanistic conclusions that spontaneous activity is not affected in the Cx30 mice. However, a quantitative comparison is missing.

To provide a more complete assessment of the potential side effects caused by the absence of connexins on the function of OHCs, we performed electrophysiological recordings from wild-type and Cx30^{-/-} P2 mice. Electrophysiology also allows a more precise quantification of the different biophysical characteristics of OHCs. As shown in the new **Supplementary Fig 6**, all parameters investigated (size and shape of the K⁺ current, ability to fire action potentials and resting membrane potential) were found to be not significantly different between the two genotypes. These results demonstrate that the absence of connexins does not affect the physiology of early postnatal OHCs, including their spontaneous action potential activity. This is now added into the legend of **Supplementary Fig 6** and in the revised main text (**ln. 240-243**).

5. Experiments with Deiters cells removed where done differently from the rest of the experiments, i.e. induction of the Ca²⁺ event by photodamage versus spontaneous Ca²⁺ waves. Why? Do both approaches examine the same process? At least the hair cells close to the damage site appear to exhibit quite distinct Ca²⁺ signals, i.e. slow and graded Ca²⁺ elevation rather than spikes (Fig. 5B).

As now mentioned in the revised MS, the main reason was to have control over when and where to generate a large calcium wave, which made the pharmacological experiments easier to perform. However, we have provided a new figure (**Supplementary Fig 4, ln. 164-168**) showing that spontaneous calcium waves elicit similar responses in OHCs as those induced by photo-damage.

6. What is the quantitative effect of Deiters cell removal on spontaneous OHC spiking (only anecdotic data shown, Suppl. fig. 2A)?

This is a similar question to that raised by Reviewer 1 (see above). In order to understand whether the biophysical properties of OHCs (including spiking activity) change in the absence of Deiters' cells (DCs), we performed additional electrophysiological recordings since, as explained above, to assess the OHC function and their health status. Recordings were done from OHCs in the absence or presence of all three rows of DCs (new **Supplementary Fig 3A-G**); in the latter, the patch pipette was advanced through the tissue with minimal positive pressure in order to minimize any disturbance. The results show that all parameters investigated (size and shape of the K^+ current, ability to fire action potentials and resting membrane potential) were not significantly different between the two recording conditions. This is now added into the revised text (**ln. 158-162**).

7. Quantification/statistics are missing for Data shown in Fig. 2 (Ca^{2+} -free, Cav1.3. ko, ryanodine)

First of all, this figure is now moved into the supplementary file (**Supplementary Fig 1**) in order to make space for the new **Fig 6**. As requested, we have included quantification for the calcium-free (**Supplementary Fig 1C**) and ryanodine (legend of **Supplementary Fig 1E**) experiments. For the Cav1.3 KO mice, we stated that we never observed calcium spikes in any of the preparations/mice investigated (legend of **Supplementary Fig 1D**).

8. Data shown in Fig. 4 lack statistics and demonstration of reproducibility (how many observations, etc.)

For **Fig 3A,B** (figure 4 in the originally submitted MS), quantification is now reported (**ln. 145-148**). **Fig 3C** provides a qualitative comparison between an electrical (Deiters' cells) and imaging (calcium wave from the GER) signal. Because of the very different nature of these two signals, especially due to the faster nature of electrical compared to Ca^{2+} signals, it is difficult to provide a reliable quantification. Therefore, we provide a quantification for the delay between the calcium signals originating in the GER and that occurring in Deiters' cells (**Fig 3D**) in the revised text (**ln. 152-154**).

Thank you for submitting a revised version of your manuscript and please accept my apologies for the delay in getting back to you with our decision. Your study has now been seen by the original referees, whose comments are shown below.

As you will see, while referee #1 finds that all criticisms have been sufficiently addressed, referee #2 still points to two unresolved issues. In particular, s/he states that the increase of OHC activity during GER Ca²⁺ waves has to be better described and carefully quantified. Also, this referee asks you to further discuss the temporal relation between GER Ca²⁺ and OHC activity and to clarify why signal onset in Deiter's cells can be observed even while the GER signaling is declining.

In addition to resolving these remaining points from referee #2, there are a few editorial issues concerning text that I need you to address before we can officially accept the manuscript.

 REFEREE REPORTS:

Referee #1:

I am satisfied with the authors' responses.

Referee #2:

The revised manuscript was substantially improved, in particular by including additional experimental evidence for the instructive role of OHC activity for shaping the afferent type II innervation pattern (Cav1.3 ko data) and for the transmission via purinergic signaling from Deiter's cells to OHCs.

Still, two major issues remain that need to be resolved.

1. Temporal coordination versus increase of OHC Ca²⁺ spike activity:

Quite obviously, the nonsensory Ca²⁺ waves both promote synchronization between OHCs' Ca²⁺ activity (and hence release) and strongly increase this activity (and hence synaptic release onto type II fibers). As already stated previously, both changes may be important for the effects on innervation.

The mechanistic argument put forward in the author's rebuttal is fully consistent and synchronization is likely to be critical. Nevertheless, the increase in OHC Ca²⁺ signals may also be essential. Thus, it is entirely possible that synchronized activity at the level observed in the absence of waves would be sufficient to drive APs in the type II fibers, but not provide enough activity for shaping the developmental process.

Consequently, it is mandatory to not only focus on one property of the response but also describe and carefully quantify the increase of OHC activity during GER Ca²⁺ waves. This should be added to Figures 2 and 7, including correlation of signal increase with Ca²⁺ wave extension and amplitude.

Along these lines, the conclusions should be modified accordingly. In particular, the following statements are at least partially misleading, as they suggest that the effect of the Ca²⁺ wave activity was to synchronize the OHC activity at baseline level without changing the activity per se.

Abstract, l. 32 "act, via ATP-induced activation of P2X3 receptors, to synchronize OHC firing, resulting in the refinement of their afferent innervation"

Introduction:, l. 68 "These spikes can be modulated by Ca²⁺ waves travelling amongst non-sensory cells via the ATP-dependent activation of P2X3 receptors, the aim of which is to synchronize the activity of nearby OHCs."

l. 71 "We propose that precisely modulated spontaneous Ca²⁺ signals between OHCs and non-sensory cells are necessary..."

Results, l. 126 "Uncorrelated spontaneous bursts of Ca²⁺ activity in nearby OHCs became highly correlated during large Ca²⁺ waves"

2. Temporal relation between GER Ca²⁺ waves and OHC activity

The overall model derived here is that activity originates in the GER, is then transmitted to the Deiter's cells, which in turn activate/modulate OHCs. In principle this should be reflected in the relative timing of the signals recorded from these structures, which is indeed supported by the example traces given in Fig. 3D.

New data quantifying a delay between GER to DC were added (l.155). However, this delay does not consistently appear in the more extensive recording shown in Fig. S2 B. In fact, DC signal onset can even be observed while the GER signal is declining in some events. This needs to be resolved (What is the meaning of the blue shaded regions in this figure? They are hard to reconcile with the onset of the rising GER signals).

Minor points:

3. New important data from Cav1.3 ko mice on fiber endings, now presented as Suppl. Fig. S8, should definitely be included in Fig. 9.

4. Fig. 10F. difficult to see which fibers are marked by asterisks (at least in printouts)

5. l.111 "Calcium waves from non-sensory cells coordinate OHC electrical activity"
similarly: l.143

Synchronization of electrical activity has not been analyzed. Better refer to Ca²⁺ signals.

6. l.131: 'spiking' is somewhat ambiguous as it could refer to electrical APs or the Ca²⁺ signals examined here.

7. l.137: "showed a positive linear relationship with the longitudinal (i.e. along the tonotopic axis) extension of Ca²⁺ waves"

While linear regression is useful to show the positive correlation, it seems to be a stretch to claim that relationship is linear, given the large variance of the data.

Ceriani et al. 'Coordinated Ca²⁺ activity of cochlear sensory and non-sensory cells refines OHC afferent innervation' – submission of the revised version of manuscript EMBOJ-2018-99839R.

Summary of major changes in response to the Reviewers Comments

We thank Reviewer 2 for his/her additional constructive remarks, which we have fully addressed in the revised MS as follows:

1) Performed additional analysis to address the issue regarding the increased OHC activity during the Ca²⁺ waves in the GER. This new data is shown in the new **Figure 2G,H** and **Appendix Fig S 2A,B** for wild-type OHCs and **Figure 7D** and **Appendix Fig S 2C,D** for Cx30^{-/-} OHCs. We have also implemented changes in the text (throughout the MS) highlighting that the increased Ca²⁺ signals in OHCs and their synchronization during Ca²⁺ waves contribute to the refinement of afferent innervation.

2) We have also clarified the second point raised by the Reviewer, which was mainly dictated by the lack of detail in the description of the previously submitted **Appendix Fig S 2** (now **Appendix Fig S 3**).

Referee #1:

I am satisfied with the authors' responses.

Thank you.

Referee #2:

The revised manuscript was substantially improved, in particular by including additional experimental evidence for the instructive role of OHC activity for shaping the afferent type II innervation pattern (Cav1.3 ko data) and for the transmission via purinergic signaling from Deiter's cells to OHCs.

Thank you.

Still, two major issues remain that need to be resolved.

1. Temporal coordination versus increase of OHC Ca²⁺ spike activity:

Quite obviously, the nonsensory Ca²⁺ waves both promote synchronization between OHCs' Ca²⁺ activity (and hence release) and strongly increase this activity (and hence synaptic release onto type II fibers). As already stated previously, both changes may be important for the effects on innervation.

The mechanistic argument put forward in the author's rebuttal is fully consistent and synchronization is likely to be critical. Nevertheless, the increase in OHC Ca²⁺ signals may also be essential. Thus, it is entirely possible that synchronized activity at the level observed in the absence of waves would be sufficient to drive APs in the type II fibers, but not provide enough activity for shaping the developmental process.

Consequently, it is mandatory to not only focus on one property of the response but also describe and carefully quantify the increase of OHC activity during GER Ca²⁺ waves. This should be added to Figures 2 and 7, including correlation of signal increase with Ca²⁺ wave extension and amplitude.

As requested, we have performed additional analysis to quantify the increase of OHC Ca²⁺ activity as a function of the extension and amplitude of the Ca²⁺ waves from non-sensory cells. This has resulted in the addition of: **Fig 2 G,H** (extension of Ca²⁺ wave described in ln. 191-211) and **Appendix Fig S 2A,B** (amplitude of Ca²⁺ wave) for wild-type OHCs; and **Figure 7D** (extension of Ca²⁺ wave: ln. 346-353) and **Appendix Fig S 2C,D** (amplitude of Ca²⁺ wave) for Cx30^{-/-} OHCs. Because of space constraints in Fig 2, we decided to move the data referring to the amplitude of the Ca²⁺ wave into the new **Appendix Fig S 2**. This new analysis shows that the Ca²⁺ signals in OHCs

increase as a function of the extension, but not amplitude, of the Ca^{2+} waves. This result is now reflected in the revised text through the MS.

One final remark: the following statement from this reviewer “*it is entirely possible that synchronized activity at the level observed in the absence of waves would be sufficient to drive*” is not correct. As shown in Figure 2, there is very little or no OHC synchronization in the absence or even in the presence of small Ca^{2+} waves.

Along these lines, the conclusions should be modified accordingly. In particular, the following statements are at least partially misleading, as they suggest that the effect of the Ca^{2+} wave activity was to synchronize the OHC activity at baseline level without changing the activity per se.

This is correct. See below for the specific statements.

Abstract, l. 32 "act, via ATP-induced activation of P2X3 receptors, to synchronize OHC firing, resulting in the refinement of their afferent innervation"

Statement changed (ln. 48-51)

Introduction:, l. 68 "These spikes can be modulated by Ca^{2+} waves travelling amongst non-sensory cells via the ATP-dependent activation of P2X3 receptors, the aim of which is to synchronize the activity of nearby OHCs."

l. 71 "We propose that precisely modulated spontaneous Ca^{2+} signals between OHCs and non-sensory cells are necessary..."

Statement changed (ln. 92-94)

Results, l. 126 "Uncorrelated spontaneous bursts of Ca^{2+} activity in nearby OHCs became highly correlated during large Ca^{2+} waves"

Statement changed (ln. 174-178)

2. Temporal relation between GER Ca^{2+} waves and OHC activity

The overall model derived here is that activity originates in the GER, is then transmitted to the Deiter's cells, which in turn activate/modulate OHCs. In principle this should be reflected in the relative timing of the signals recorded from these structures, which is indeed supported by the example traces given in Fig. 3D.

New data quantifying a delay between GER to DC were added (l.155). However, this delay does not consistently appear in the more extensive recording shown in Fig. S2 B. In fact, DC signal onset can even be observed while the GER signal is declining in some events. This needs to be resolved (What is the meaning of the blue shaded regions in this figure? They are hard to reconcile with the onset of the rising GER signals).

We thank the Reviewer for pointing out this apparent inconsistency, which was due to a lack of detail in the description of **Appendix Fig S 2** (now **Appendix Fig S 3**). We have now expanded the Figure legend to provide more information about the analysis used and the intended meaning of this Supplementary Figure, and how it differs to that in **Fig. 3D**.

Appendix Fig S 3 is used to provide a qualitative demonstration that repetitive Ca^{2+} signals in the GER (red) can be mirrored in the Deiters' cells (blue and green). The GER trace (red) was obtained by integrating the Ca^{2+} signal over the entire area of the Ca^{2+} wave (red square in the left panel), and as such containing signals propagating among non-sensory cells in all directions (e.g. orthogonal and along the coiling axis of the cochlea). This is also evident by the much “smoother” GER trace compared to the Deiters' trace (blue), which was from a single cell. Because of the different analysis used for the GER (red) and the Deiters' trace (blue) the latter “appears” to begin to return to the baseline fluorescence before those from the GER. This is now explained in the Figure legend of **Appendix Fig S 3**.

Different from **Appendix Fig S 3**, the quantification provided in **Fig. 3D** was obtained by measuring Ca^{2+} signals from single non-sensory cells in the GER and Deiters' cell. We now state this in the legend of Figure 3 (**ln. 1087-1089**).

The blue shaded regions have been removed to avoid any confusion.

Minor points:

3. New important data from Cav1.3 ko mice on fiber endings, now presented as Suppl. Fig. S8, should definitely be included in Fig. 9.

As requested, we have moved the OHCs data from *Cav1.3 KO* mice to the Revised Figure 9. In order to make space for the OHC data, we have moved the IHC results, which are less relevant to the paper, into the Revised **Appendix Fig S 8**.

4. Fig. 10F. difficult to see which fibers are marked by asterisks (at least in printouts)
The asterisks have been moved closer to the fibres in the revised **Fig. 10**.

5. 1.111 "Calcium waves from non-sensory cells coordinate OHC electrical activity"
similarly: 1.143

Synchronization of electrical activity has not been analyzed. Better refer to Ca^{2+} signals.

As suggested, we have changed the term "electrical activity" to either " Ca^{2+} activity" or " Ca^{2+} signals" throughout the MS, apart when referring to the activity recorded using electrophysiological experiments.

6. 1.131: 'spiking' is somewhat ambiguous as it could refer to electrical APs or the Ca^{2+} signals examined here.

As suggested, the word spiking has been changed to Ca^{2+} signals throughout the MS, unless it was related to electrical APs.

7. 1.137: "showed a positive linear relationship with the longitudinal (i.e. along the tonotopic axis) extension of Ca^{2+} waves"

While linear regression is useful to show the positive correlation, it seems to be a stretch to claim that relationship is linear, given the large variance of the data.

The word "linear" has been removed.

Corresponding Author Name: Walter Marcotti

Manuscript Number: EMBOJ-2018-99839R 1